# Discovering Mathematical Formulas from Data via LSTM-guided Monte Carlo Tree Search

## Abstract

Finding a concise and interpretable mathematical formula that accurately describes the relationship between each variable and the predicted value in the data is a crucial task in scientific research, as well as a significant challenge in artificial intelligence. This problem is commonly referred to as symbolic regression, which poses an NP-hard combinatorial optimization problem. Traditional symbolic regression algorithms typically rely on genetic algorithms; however, these approaches are sensitive to hyperparameters and often struggle to fully recover the target expression. To address these limitations, a novel symbolic regression algorithm based on Monte Carlo Tree Search (MCTS) was proposed this year. While this algorithm has shown considerable improvement in recovering target expressions compared to previous methods, it still faces challenges when dealing with complex expressions due to the vast search space involved. Moreover, the lack of guidance during the MCTS expansion process severely hampers its search efficiency. Recently, some algorithms add a pre-trained policy network to guide the search of MCTS, but the pre-trained policy network generalizes poorly. In order to overcome these issues, we propose AlphaSymbol combining ideas from AlphaZero. AlphaSymbol is a new symbolic regression algorithm that combines MCTS with a Long Short-Term Memory network (LSTM). By leveraging LSTM's ability to guide MCTS expansion process effectively, we enhance the overall search efficiency of MCTS significantly. Next, we utilize the MCTS results to further refine the LSTM network, enhancing its capabilities and providing more accurate guidance for the MCTS process. MCTS and LSTM are coupled together and optimize each other until the target expression is successfully determined. We conducted extensive evaluations of AlphaSymbol using 222 expressions sourced from over 10 different symbolic regression datasets. The experimental results demonstrate that AlphaSymbol outperforms existing state-of-the-art algorithms in accurately recovering symbolic expressions both with and without added noise.

## 1 Introduction

The ultimate goal of natural scientific research is to discover a concise and clear mathematical expression from a set of data, which can describe the relationship between variables in the data and reflect the objective laws of the physical world. The goal of symbolic regression is to use observed data to search for an expression that can fit the data well. Specifically, if there is a set of data $[x_1, x_2, ..., x_m, y]$ where $x_i \in \mathbb{R}^n$ and $y \in \mathbb{R}$, the purpose of symbolic regression is to discover a mathematical expression $f(x_1, x_2, ..., x_m)$ through certain methods so that $f$ can fit the data $y$ well. The resulting expression can not only fit the data y well but also be interpretable. We can utilize the properties of the basic operators in the expression $f$ to analyze the relationship between the feature variables $[x_1, x_2, ..., x_m]$ and $y$ in the data. In recent years, deep learning has penetrated various areas of our lives(Chen et al., 2022; Choudhary et al., 2022; Zhang et al., 2021). People from many fields, from physics to life sciences, are using neural networks to fit their data. Although artificial neural networks can fit the data well, the result obtained by the neural network fitting is a "black box"(Petch et al., 2022), which is not interpretable and analyzable. On the contrary, the result of symbolic regression is a clear, interpretable, and analyzable mathematical expression. For example,

in the physical formula $\mathcal{P} = \mathcal{FV}$, we can easily analyze that, given a constant power $\mathcal{P}$, to obtain a larger force $\mathcal{F}$, we must decrease the velocity $\mathcal{V}$. This is also why vehicles slow down when going uphill in real-life scenarios. However, a black-box model obtained from a neural network cannot intuitively derive such useful conclusions.

Mathematical expressions are composed of basic operators, and any expression can be expressed as an expression binary tree. If we expand the expression binary tree in the order of preorder traversal, we can obtain an ordered and discrete sequence of operators. So essentially symbolic regression can be regarded as a combinatorial optimization problem (Karimi-Mamaghan et al., 2022). This is an NP-hard problem (Huynh et al., 2022). Traditional approaches to symbolic regression typically utilize evolutionary algorithms, especially genetic programming (GP)(Koza et al., 1992; Schmidt & Lipson, 2009; Haarnoja et al., 2018). In GP-based symbolic regression, the algorithm begins by initializing a population of expressions. The individuals in the population undergo crossover and mutation operations to simulate human evolution. Finally, the best individuals are selected to form the next generation population. While GP can be effective, it is also known to scale poorly to larger problems and exhibit high sensitivity to hyperparameters.

Symbolic Physics Learner (SPL)(Sun et al., 2022), published this year, employs MCTS to address symbolic regression problems. In SPL, MCTS iteratively executes four steps (selection, expansion, simulation, and backtracking) to progressively uncover the optimal search path. This paradigm effectively tackles symbolic regression problems and demonstrates remarkable performance; Appendix I delineates a succinct example of employing MCTS to tackle the symbolic regression. however, due to completely random selection during initial expansion and throughout the simulation phase without any guidance mechanism in place results in suboptimal search efficiency for SPL. The DGSR-MCTS(Kamienny et al., 2023) and TPSR(Shojaee et al., 2023) algorithm integrates a pre-trained policy network to facilitate the search mechanism of the MCTS. However, this pre-trained model demonstrates limited generalizability, exhibiting satisfactory performance only on data configurations encountered during its training phase. For instance, should the training be confined to a symbolic repository encompassing $[+, sin, cos, x]$, any subsequent attempts to extend the repository to incorporate additional symbols such as $[+, -, sin, cos, exp, x]$ results in suboptimal performance, or may altogether fail to function. Furthermore, the model's efficacy is significantly compromised when exposed to scenarios deviating from its training conditions; a model trained exclusively with two variables, $x_1$ and $x_2$, is markedly less competent when evaluated on datasets featuring three variables. Even when the model samples X within the interval [-2, 2] during training, its performance is greatly compromised when we sample X within the range of [-4, 4] for testing. In light of these considerations, both generalization and search efficiency of the algorithm must be addressed. To this end, we introduce an innovative algorithm, termed AlphaSymbol, designed to reconcile the two. Inspired by the AlphaZero(Silver et al., 2018) algorithm. We propose $AlphaSymbol$, a novel symbolic regression framework based on LSTM and MCTS. In $AlphaSymbol$, we integrate the policy network and value network into one. We employ an LSTM to generate the probability $p$ of selecting each symbol and the value $v$ of the current state, which guides the MCTS during the expansion and simulation phases. Before the self-search actually generates a symbol, multiple simulations are conducted. Once the self-search generates a complete expression. we calculate the reward using the reward function and perform backpropagation. Furthermore, we collect the node information selected during the self-search process to train the LSTM neural network. We collect the parent node and sibling node of the current node as the input of the LSTM network and take the normalized value $\pi$ of the number of times it is child nodes have been visited (probability value of the child nodes being selected) and the state value $z$ as output. Then, using these data, we can train the LSTM network to become even more powerful, which can better guide the MCTS algorithm. We summarize our contributions as follows:

- We propose a novel symbol regression model, called AlphaSymbol, which cleverly combines LSTM and MCTS. And outperforms several baselines on a series of benchmark problems. Source code is provided at [1].

- We propose a new loss function (Used to calculate rewards) called $S_{NRMSE}$, which effectively addresses the issue of variable omission in multivariate regression problems.

- We improve the loss function (used to train LSTM) to encourage the LSTM to produce a probability distribution with lower information entropy, thereby avoiding situations where

---

[1]Source code for AlphaSymbol: `https://anonymous.4open.science/r/AlphaSymbol-v2`

each symbol is predicted with a similar probability. Improved the search efficiency of the algorithm.

## 2 RELATED WORK

**Deep learning for symbolic regression**. Recently, many algorithms have been developed to apply deep learning to symbol regression, achieving promising results. EQL(Martius & Lampert, 2016; Kim et al., 2020) replaces the activation function of neural networks with basic operators such as $[+, -, *, /, sin, ...]$, Afterwards, parameter sparsification is conducted to eliminate redundant connections and extract expressions from the network. AI Feynman consists of two versions, AI Feynman 1.0(Udrescu & Tegmark, 2020) and AI Feynman 2.0(Udrescu et al., 2020), both of which aim to simplify complex problems. In AI Feynman 1.0, a neural network is first trained to fit data and then used to discover some properties, such as additivity separability, Then, these properties are used to break down a complex problem into several simpler ones. The limitation is that AI Feynman 1.0 applies only a limited number of properties and achieves better performance only in the domain of physics expressions. Building upon the foundation of AI Feynman 1.0, version 2.0 proposes more properties, expanding the algorithm's applicability to any field. The NeSymRes(Biggio et al., 2021) algorithm treats symbolic regression as a machine translation problem. This algorithm trains a transformer model with a large amount of data and then uses the model with beam search to generate expressions. This article(Li et al.) replaces the feature extraction module of NeSymRes with the point cloud feature extraction algorithm pointMLP.

**Genetic programming for symbolic regression**. Genetic Algorithm (GA)(Zhong et al., 2015; Huang et al., 2022; Haider et al., 2023) is a classic optimization algorithm that simulates the evolution of human history. The symbolic regression algorithm based on GA is Genetic programming (GP). GP first initializes a population of expression binary trees. Then, a simulation of human "evolution" is carried out through means such as crossover and mutation. Finally, excellent individuals are selected as the next-generation population through fitness selection. This process is repeated. In addition, there are many algorithms that have been developed by improving upon the GP algorithm(Zhong et al., 2015; Huang et al., 2022; Haider et al., 2023).

**Reinforcement learning for symbolic regression**. DSR(Petersen et al., 2019) and DSO (Mundhenk et al., 2021) are two excellent symbolic regression algorithms based on deep reinforcement learning. Among them, DSR defines Recurrent Neural Networks (RNN(Graves, 2013), LSTM(Gers et al., 2000), GRU (Chung et al., 2014), etc) as the policy network. The network takes the parent and sibling nodes of the node to be generated as inputs and outputs the probability of selecting each symbol. Then, multiple expressions are sampled from the policy network. Finally, the reward values are calculated, and the policy network parameters are updated through policy gradient to enable the policy network to generate better expressions. DSO introduces the GP on the basis of DSR. In DSO, the expressions sampled from the policy network are used as the initial population for the GP. Then the GP-evolved expressions and original expressions are combined to update the policy network in order to provide higher-quality initial populations for GP. SPL(Sun et al., 2022) applies the successful conventional MCTS to the symbolic regression field. The algorithm selects a good sequence of expressions by repeating four steps: selection, expansion, simulation, and backpropagation.

## 3 MODELING

The symbol library contains a series of basic operators that can be flexibly combined into various mathematical expressions. Our symbol library includes five binary operators $[+, -, \times, \div, \bullet^\bullet]$, five unary operators $[\sin, \cos, \exp, \sqrt{\ }, \ln]$, multiple variable operators $[x_1, x_2, \ldots, x_n]$, as well as constant placeholders $[c]$. Guided by an LSTM, the AlphaSymbol executes MCTS, generating operators for the expression in the order of $pre - order$ traversal of the expression's binary tree. AlphaSymbol uses an LSTM $\mathcal{N}_\theta$ with parameter $\theta$. This LSTM combines the roles of both the policy network and value network into a single architecture. $\mathcal{N}_\theta$ takes as an input the parent node and sibling nodes of the node to be predicted and outputs the probability $p$ of selecting each operator and the current state value $v$, $(p, v) = \mathcal{N}_\theta(s)$. Where the state value $v$ can be seen as the degree to which continuing downward selection from the current node can eventually lead to the optimal expression. The LSTM in AlphaSymbol is trained from the task of self-search by a reinforcement learning algorithm. In self-search, before selecting a new symbol, we will conduct several simulations

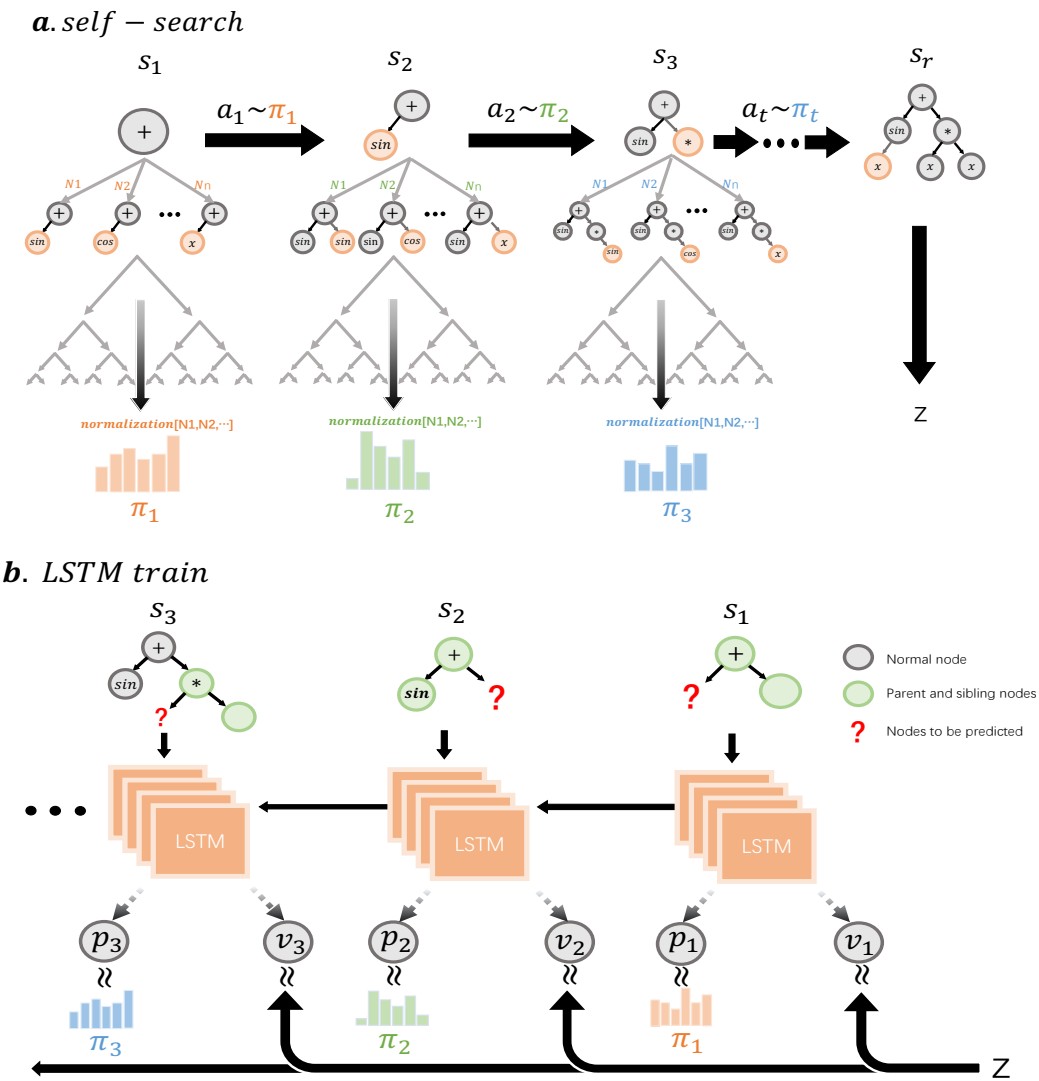

Figure 1: **a. self-search.** During the self-search phase, the operators with the highest probabilities (Number of visits) are selected sequentially for each state $s_1, s_2, ..., s_t$. For each state $s_t$, multiple simulations are carried out, and during each simulation, an MCTS $\alpha_\theta$ using the latest neural network $\mathcal{N}_\theta$ is executed (as depicted in Fig 2). Finally, by tabulating the number of times each child node was visited during multiple simulations, we can determine the selection probability $a_t \sim \pi_t$ for each state. Finally, when the expression is complete at time $s_r$, we calculate the reward value $z$ and perform backpropagation. **b. LSTM training.** In AlphaSymbol, the LSTM is designed to take the state $s_t$ as input, which is then passed through an LSTM with parameters $\theta$. The output comprises a vector $p_t$ and a scalar value $v_t$, where $p_t$ represents a probability distribution over candidate symbols and $v_t$ represents the possible reward value after generating a complete expression starting from the current state $s_t$. During training, the LSTM parameters, $\theta$, are updated to maximize the similarity between the policy vector, $p_t$, and the search probabilities, $\pi_t$, while minimizing the difference in the predicted reward $v_t$ and the actual reward $z$. The new parameters are used in the next iteration of self-search as in Fig (a).

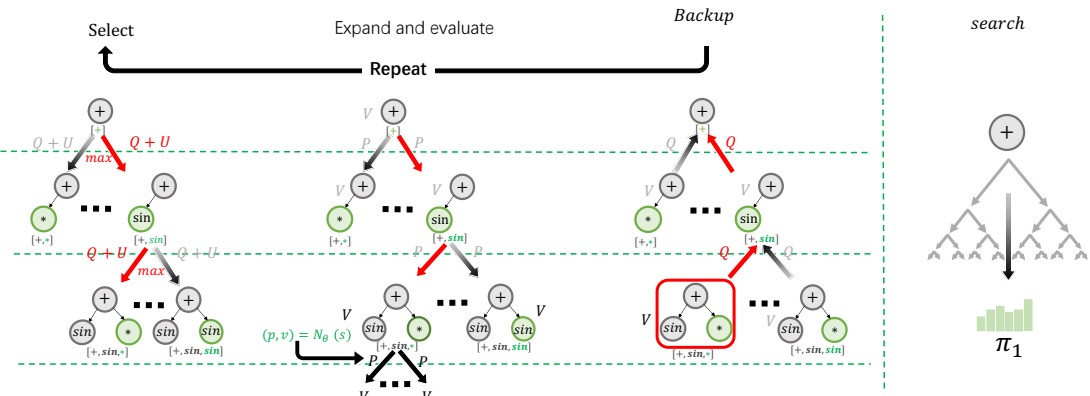

Figure 2: **MCTS in AlphaSymbol**. **Select,** Starting from the root node, the child node with the largest UCT value is selected recursively, where $\mathcal{UCT} = \mathcal{Q}(s,a) + \mathcal{U}(s,a)$ and finally a leaf node is reached. **Expand,** If the current leaf is not a terminal node, then create multiple (number of symbols) child nodes, and select the one with the greatest probability $p(s,a)$ to expand., where $p(s,\bullet) = \mathcal{N}_\theta(s)$, and the $p(s,a)$ values are stored in the corresponding new child nodes. **evaluate,** Use LSTM to calculate the $V$ of the new extension node. **backpropagation,** After each simulation, backpropagation is performed, where $\mathcal{N}_{visit}$ of all visited nodes during the simulation is incremented by one. Additionally, Action value $Q$ is updated to track the mean of all evaluations V in the subtree below that action. **Search,** The probability $\pi$ of selecting each symbol can be determined during the self-search phase after the completion of the simulation phase. Herein, $\pi$ is directly proportional to $(\mathcal{N}_{visit})^{\frac{1}{\tau}}$, where $\mathcal{N}_{visit}$ is the visit count of each move from the root state and $\tau$ is a parameter controlling temperature.

under the guidance of LSTM $\mathcal{N}_\theta$. After several MCTS simulations, the number of times $N_i$ each possible child node is selected is obtained, and then $[N_1, N_2, ...N_n]$ is normalized to obtain $\pi$. Then, based on the probability $\pi$, the program is instructed to actually choose a symbol. These search probabilities usually select much stronger moves than the raw move probabilities $p$ of the neural network $\mathcal{N}_\theta(s)$; MCTS may therefore be viewed as a powerful policy improvement operator. When we have a completed expression, we will calculate the reward $z$ through the reward function, $\mathcal{R}(s) = 1/(1 + S_{NRMSE})$, where $S_{NRMSE}$ is shown in expression 6. Repeat the process until you get the target expression. the neural network's parameters are updated to make the operator's probabilities and value $(p, v) = \mathcal{N}_\theta$ more closely match the improved search probabilities and self-search reward $(\pi, z)$; these new parameters are used in the next iteration of self-search to make the search even stronger. Figure 1 a illustrates the self-search training process.

As shown in Figure 2, MCTS is simulated several times under the guidance of an LSTM network. During the simulation, each node stores three kinds of information, prior probability $\mathcal{P}(s,a)$, a visit count $\mathcal{N}(s,a)$, and an action value $\mathcal{Q}(s,a)$. In the simulation stage, each simulation starts from the root node, and the node with the largest $\mathcal{UCT}$ (Gelly & Silver, 2007) value among the child nodes is selected successively. The $\mathcal{UCT}$ expression is shown in expression below 1:

$$\mathcal{UCT}(s,a) = \mathcal{Q}(s,a) + \mathcal{U}(s,a) \tag{1}$$

Here Action value $\mathcal{Q}(s,a)$ (Silver et al., 2017)is the cumulative average reward value of the current state, and the expression is 2:

$$\mathcal{Q}(s,a) = \frac{1}{\mathcal{N}(s,a)} \sum_{s'|s,a \to s'} \mathcal{V}(s') \tag{2}$$

And $s, a \to s'$ indicates that a simulation eventually reached $s'$ after taking move $a$ from position $s$. where $\mathcal{U}(s,a) \propto p(s,a)/(1 + \mathcal{N}(s,a)$, The specific expression is 3:

$$\mathcal{U}(s,a) = c_{puct} * \mathcal{P}(s,a) * \frac{\sqrt{\mathcal{N}(s)}}{1 + \mathcal{N}(s,a)} \tag{3}$$

When a leaf node $s'$ is encountered during the simulation, if the leaf node is not a terminal node, this leaf position is expanded and evaluated only once by the network to generate both prior probabilities

and evaluation, $(\mathcal{P}(s^{'}, \bullet), \mathcal{V}(s^{'})) = \mathcal{N}_\theta(s^{'})$. Finally, a backpropagation step is performed to update the *counter*, action value $\mathcal{Q}(s, a)$, visit count $\mathcal{N}(s, a)$, and *length* for the nodes traversed during the simulation. The self-search process of AlphaSymbol can be perceived as a symbolic selection process where symbols are chosen through a greedy search algorithm guided by a probability $\pi$. The symbol with the maximum $\pi$ value is selected as a new symbol. In addition, $\pi$ is proportional to the number of visits of each child node after multiple simulations. Here $\pi_a \propto \mathcal{N}(s, a)^{\frac{1}{\tau}}$, and its precise computational expression is as follows 4.

$$\pi_{a_i} = \frac{log(\mathcal{N}(s, a_i)^{\frac{1}{\tau}})}{\sum_{i=0}^{n} log(\mathcal{N}(s, a_i)^{\frac{1}{\tau}})} \tag{4}$$

The training of LSTM is a regression problem. Firstly, we initialize the state as $s_0$, and the parameters of the LSTM network are randomly initialized as $\theta_0$. During the self-search phase, before making an actual selection of a symbol, we conduct multiple MCTS simulations under the guidance of the LSTM network, with the current state $s_0$ as the input. Then, we obtain the probability $\pi_0$ and select symbol $a_0$ to move into state $s_1$ under the guidance of $\pi_0$. Train and update the parameters of the LSTM using the data collected during the self-search phase to obtain $\theta_1$. Repeat the above process until a complete expression is obtained or the maximum length is reached. Once a complete expression is obtained, we compute the final reward $z$ and perform backpropagation. Each traversed node during the self-search process can contribute to training data $(s_t, \pi_t, z_t)$, where $z_t = z$. Figure 1 b illustrates that the LSTM neural network is trained by randomly selecting a portion of the previously collected training data $(s, \pi, z)$, which generates a new set of parameters designated as $\theta_{new}$. The LSTM neural network $(p, v) = \mathcal{N}_{\theta_{new}}(s)$ is adjusted to minimize the error between the predicted value $v$ and the reward $z$ and to maximize the similarity of the neural network predicts probabilities $p$ to the search probabilities $\pi$. Specifically, In this study, the parameters $\theta$ of the LSTM are updated via the gradient descent method. Expression 5 shows the loss function to be optimized. It is noteworthy that we incorporate the information entropy of the predicted probability $p$ as a part of the loss function.

$$\mathcal{L} = (z - v)^2 - \pi^T logp - p^T logp + \xi||\theta||^2 \tag{5}$$

Among them, the first term $(z - v)^2$ makes the predicted value $v$ and the true value $z$ as close as possible. The second term, $-\pi^T \log p$, minimizes the difference between predicted probability $p$ and $\pi$. The third term, $-p^T \log p$(Wulfmeier et al., 2015), the information entropy loss which maximizes the difference between the predicted probability of each symbol so that the probabilities are concentrated on a single symbol rather than being almost equal for every symbol. The final term, $\xi||\theta||^2$, is the $L_2$ (Van Laarhoven, 2017) regularization penalty to prevent overfitting of the neural network, where $\xi$ is a parameter that controls the strength of $L_2$ weight regularization. The pseudocode for AlphaSymbol is shown in Algorithm 1

**Constant optimization**. If a constant placeholder "C" appears in the sampled expression, we will use nonlinear optimization algorithms such as BFGS (Liu & Nocedal, 1989) or L-BFGS-B (Zhu et al., 1997) to optimize the constants. For example, for the expression $2.2 * sin(x) + 1.3$, we might search for an expression of the form $C * sin(x) + C$, and then perform constant optimization using BFGS to obtain the original expression.

**Loss function :** $S_{NRMSE}$. Traditional loss functions only calculate loss by evaluating the match between predicted value $\hat{y}$ and actual value $y$. The expression for the loss is $NRMSE = \frac{1}{\sigma}\sqrt{\frac{1}{N}\sum_{i=1}^{N}(y_i - \hat{y}_i)^2}$(Chai & Draxler, 2014). When dealing with multivariate problems, especially when one variable is numerically much smaller than the others, or when two variables are highly correlated, using NRMSE as a loss metric may easily result in variable omission. To tackle this challenge, we propose a new loss function called "$S_{NRMSE}$," with the following expression:6

$$\mathcal{S_{NRMSE}} = \frac{1}{\sigma_y}\sqrt{\frac{1}{\mathcal{N}}\sum_{i=1}^{\mathcal{N}}(y_i - \hat{y}_i)^2} + \lambda * \sum_{j=1}^{m}\frac{1}{\sigma_{x_j}}\sqrt{\frac{1}{\mathcal{N}}\sum_{i=1}^{\mathcal{N}}(x_{ji} - \hat{x}_{ji})^2} \tag{6}$$

Here, $\mathcal{N}$ represents the sample size, $m$ represents the number of variables, and $x_{ji}$ refers to the $i^{th}$ variable of the $j^{th}$ sample. $\lambda$ belongs to the interval $[-1, 1]$, indicating the importance of $\mathcal{X}$ in the loss function. if there are variables $[x_1, x_2, ..., x_j]$ in the predicted expression, then we simply set the predicted value $[\hat{x_1}, \hat{x_2}, ..., \hat{x_j}]$ of these variables to their original value, and the loss is 0. If some variables $[x_1, x_2, ..., x_j]$ are missing from the prediction expression, we set the predicted value of

Table 1: Recovery rate comparison of AlphaSymbol and four baselines on more than ten mainstream symbolic regression datasets.

| Dataset | Dataset | Number | AlphaSymbol | DSO | SPL | GP | NeSymReS |
|---|---|---|---|---|---|---|---|
| Dataset-1 | Nguyen | 21 | $\mathbf{95}_{\pm1.49}\%$ | $92_{\pm2.68}\%$ | $91_{\pm3.46}\%$ | $61_{\pm4.17}\%$ | $56_{\pm3.67}\%$ |
| Dataset-2 | Keijzer | 15 | $78_{\pm2.81}\%$ | $\mathbf{81}_{\pm3.17}\%$ | $76_{\pm3.24}\%$ | $47_{\pm6.33}\%$ | $52_{\pm5.93}\%$ |
| Dataset-3 | Korns | 15 | $\mathbf{74}_{\pm1.89}\%$ | $73_{\pm3.32}\%$ | $69_{\pm4.10}\%$ | $31_{\pm5.21}\%$ | $27_{\pm5.29}\%$ |
| Dataset-4 | Constant | 8 | $\mathbf{88}_{\pm3.14}\%$ | $85_{\pm4.89}\%$ | $82_{\pm4.23}\%$ | $35_{\pm5.14}\%$ | $22_{\pm5.17}\%$ |
| Dataset-5 | Livermore | 22 | $\mathbf{89}_{\pm1.67}\%$ | $85_{\pm2.26}\%$ | $81_{\pm2.46}\%$ | $40_{\pm3.17}\%$ | $28_{\pm3.58}\%$ |
| Dataset-6 | Vladislavleva | 8 | $\mathbf{52}_{\pm2.12}\%$ | $48_{\pm2.95}\%$ | $39_{\pm3.29}\%$ | $12_{\pm2.47}\%$ | $13_{\pm3.20}\%$ |
| Dataset-7 | R | 6 | $\mathbf{46}_{\pm1.28}\%$ | $44_{\pm1.94}\%$ | $32_{\pm2.61}\%$ | $9_{\pm2.24}\%$ | $4_{\pm3.18}\%$ |
| Dataset-8 | Jin | 6 | $\mathbf{60}_{\pm1.73}\%$ | $53_{\pm2.10}\%$ | $43_{\pm2.41}\%$ | $14_{\pm2.19}\%$ | $11_{\pm3.01}\%$ |
| Dataset-9 | Neat | 9 | $67_{\pm2.01}\%$ | $\mathbf{72}_{\pm2.48}\%$ | $65_{\pm2.47}\%$ | $27_{\pm3.08}\%$ | $21_{\pm3.15}\%$ |
| Dataset-10 | AIFeynman | 103 | $\mathbf{64}_{\pm3.02}\%$ | $56_{\pm3.85}\%$ | $44_{\pm4.10}\%$ | $29_{\pm4.95}\%$ | $18_{\pm4.62}\%$ |
| Dataset-11 | Others | 9 | $\mathbf{81}_{\pm1.28}\%$ | $75_{\pm1.86}\%$ | $74_{\pm2.19}\%$ | $35_{\pm3.04}\%$ | $24_{\pm2.73}\%$ |
| | | Average | $\mathbf{72.2}_{\pm10.62}\%$ | $69.5_{\pm11.1}\%$ | $63.1_{\pm13.4}\%$ | $30.0_{\pm10.8}\%$ | $25.1_{\pm10.74}\%$ |

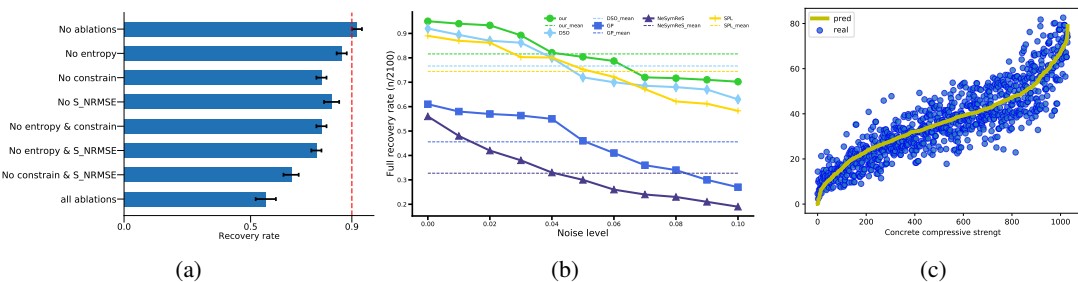

(a)           (b)           (c)

Figure 3: Fig (a) illustrates the recovery for various ablations of AlphaSymbol on all Nguyen benchmarks, with error bars indicating the standard errors. Fig (b) describes the recovery rate of AlphaSymbol and four other excellent algorithms on all Nguyen benchmarks under different levels of noise. Fig (c) describes the relationship between the fitted curve and the real data. From this, we can clearly see that the yellow curve robustly fits the sample points.

these variables to [0, 0,...,0] to generate a large loss. To achieve the purpose of preventing variable loss.

**Reward function**. When a complete expression is obtained during the self-search phase, we need to use a reward function to evaluate its quality. Firstly, the reward function must meet the requirement that the smaller loss, the larger reward. Secondly, to make the learning process more stable, the reward function should be designed to have values that range between $[0, 1]$. Therefore, we define the reward function as $Reward(s) = \frac{1}{1+S_{NRMSE}}$

**Constraining the search space**. To ensure the smooth operation of the algorithm and enhance its efficiency and performance. We have imposed some reasonable constraints on the search space of the algorithm. (1), We have placed a limit on the length of the generated expression. (2), Operators that are inverse functions cannot occur consecutively (for example, the operator $exp(log(x))$ is not allowed). (3), We have disallowed the repeated nesting of $sin$ and $cos$ because, in real-life applications, expressions with repeated nesting of trigonometric functions are rare (e.g. $sin(cos(x))$ is not allowed). (4), Some illegal expressions are not allowed to be generated. For example, $log(\tau)$, where $\tau$ should be greater than zero, so in the process of expression generation, functions that have negative values like $sin$, $cos$, and so on can't come right after $log$.

**Expression generation termination**. Introduce two variables, $counter$ and $arity$(Petersen et al., 2019), where the $counter$ is initialized to 1. Binary functions, such as $[+, -, \times, \div ...]$, have an $arity$ of 2, while unary functions, such as $[sin, cos, exp, ln...]$, have an $arity$ of 1. Variables, $[x_1, x_2, ..., x_n]$, and constants placeholders $[c]$ have an $arity$ of 0. During the process of generating expressions through pre-order traversal, the current $counter$ of the symbol sequence is calculated in real-time using the expression $counter = counter + arity(a) - 1$. If the $counter$ of the current sequence is 0, it indicates that a complete expression has been sampled, and the sampling is stopped.

## 4 RESULTS

**Evaluating AlphaSymbol**. We evaluated the performance of AlphaSymbol on more than ten classic datasets in the field of symbolic regression. These datasets are labeled $Nguyen, Keijzer, Korns, Constant, Livermore, R, Vladislavlev, Jin, Neat, AIFeynman$, and $Others$. The datasets mentioned above collectively contain a total of 222 test expressions, The specific details are shown in table tables C.1 to C.3 in the appendix. We compare AlphaSymbol with four symbol regression algorithms that have demonstrated high performance:

- **SPL**. An excellent algorithm which successfully applies the traditional MCTS to the field of symbolic regression.

- **DSO**. A superior algorithm that effectively integrates DSR and GP for symbolic regression tasks.

- **GP**. A classic algorithm that applies genetic algorithms perfectly to the field of symbolic regression.

- **NeSymReS**. This algorithm is categorized as a large-scale pre-training model.

In order to test the ability of each algorithm to fully recover expression using only local data, we sampled only 20 points for variable $X$ within the interval $[-1, 1]$ and obtained the corresponding true value $y$. We use the strictest definition of recovery: exact symbolic equivalence, as determined using a computer algebra system, e.g. SymPy (Meurer et al., 2017). We tested each expression 100 times using the aforementioned algorithms and recorded the number of times, denoted by $\mathcal{N}_{full}$, that expression was completely recovered (The expression symbols, constant positions, and constant values are all the same). Finally, we evaluated the performance of each algorithm by comparing the full recovery rate, calculated as $\mathcal{N}_{full}/100$. In Table 1, we present the recovery rates of each algorithm on all benchmark datasets. The performance of AlphaSymbol on recovery rate is slightly better than the other four advanced algorithms. In addition, the $R^2$ of AlphaSymbol on the AI Feynman dataset is presented in E.1E.2. As the training progresses, the algorithm's reward function fluctuation is illustrated in the line graph (convergence proof) as depicted in Figures B.1 of the appendix. From the reward fluctuation line chart, one can discern that as training ensues, the reward values exhibit an oscillatory ascent, eventually culminating in a state of equilibrium. This corroborates the efficacy of our algorithm in guiding symbolic search, and it also validates that our algorithm ultimately attains convergence.

**Ablation studies**. AlphaSymbol includes a series of small but important components, in addition to its main algorithm. We developed a series of ablation experiments to verify the effect of each component on the algorithm performance. The image in Fig 3a shows the AlphaSymbol performance change on all Nguyen benchmarks after the different component combinations are removed. Where "No entropy" means that the information entropy loss is not applied to the loss. "No constrain" means no constraints are applied. "No S_NRMSE" means that the $S_{NRMSE}$ loss function was not applied. As can be seen from the figure, although there is no catastrophic decline in algorithm performance after different components are removed, the recovery rate is significantly reduced compared with that without the ablation experiment.

Table 2: This demonstrates the impact of incorporating the information entropy of predicted probability $p$ into the loss function during LSTM training on the efficiency of the algorithm.

| Benchmark | Mean entropy | | Time(s) | |
|---|---|---|---|---|
| | Yes | No | Yes | No |
| Nguyen-1 | **1.50** | 2.16 | **14.2** | 25.6 |
| Nguyen-2 | **1.62** | 2.43 | **115.34** | 168.16 |
| Nguyen-3 | **1.82** | 2.96 | **268.42** | 484.76 |
| Nguyen-4 | **2.19** | 3.02 | **268.42** | 484.76 |
| Average | **1.78** | 2.64 | **132.65** | 226.17 |

**Anti-noise experiment** In the natural world, most of the data we get has some level of noise. Therefore, the anti-noise ability is an important index to test whether an algorithm can solve real problems. We conducted anti-noise experiments on the AlphaSymbol algorithm and four other

advanced algorithms on Nguyen benchmarks. The noisy data(Shang et al., 2018) were randomly sampled from the range of $[-level * scale, level * scale]$ where $level \in [0, 0.1]$ represents the noise level, and the $scale$ is equal to $max(y) - min(y)$. At each noise level, we ran each expression 100 times. Then, we calculated the recovery rate of each algorithm on the expressions at each noise level. As shown in Fig 3b, AlphaSymbol exhibits outstanding performance in anti-noise ability. It outperforms all other algorithms at all noise levels except for slightly lower performance than DSO at noise levels of 0.01 and 0.02.

**Ablation experiment for information entropy**. In order to increase the "confidence" of the LSTM in selecting the correct symbol, it is highly preferable that the predicted probability distribution $p$ be concentrated on the correct symbol. We incorporated the information entropy of the probability distribution $p$, represented as $-p^T log(p)$, into the loss function. We performed ablation experiments on three expressions Nguyen-1 to Nguyen-3. During the experimentations, the predicted values $[p_1, p_2, p_3, \dots]$ were retained after every prediction of the LSTM network. The information entropy $E_i$ for each $p_i$ was then calculated as $E_i = -p_i^T \log p_i$. Finally, the mean information entropy, $E_{mean}$, was determined by averaging the information entropy for each $p_i$. Additionally, we monitored the impact of incorporating the information entropy of the predicted probability distribution $p$ into the loss function on the algorithm's efficiency. As shown in Table 2, incorporating the information entropy of $p$ into the loss function makes the LSTM network more "confident" in its predictions, resulting in a lower information entropy of the predicted probabilities $p$. This indicates that the LSTM is more certain in selecting a symbol. Meanwhile, the LSTM network's increased "confidence" can result in improved algorithm efficiency. Under equal conditions, we are able to find the target expression more quickly.

**The resulting analyzability test** This Concrete compressive strength data set (Yeh, 1998) contains 5 quantitative input variables and 1 quantitative output variable. The 5 input variables are Cement ($x_1$), Blast Furnace Slag ($x_2$), Water ($x_3$), Superplasticizer ($x_4$), and Age ($x_5$). And the output variable is Concrete compressive strength ($y$). We use the aforementioned 5 input variables as input into AlphaSymbol to fit the output variable Concrete compressive strength. The final expression derived by AlphaSymbol is as shown in equation 7.

$$y = 0.66x_4 + \frac{0.66(x_1 + x_2 - x_3 + 334)}{7.17 + \frac{17746.56}{x1*x5}} \tag{7}$$

We can analyze the positive or negative correlation between the variables $[x1, x2, ..., x_5]$ and $y$ very clearly from these equations. According to Formula 8, we can take the partial derivative of $y$ with respect to $x$, where $\frac{\partial y}{\partial x_1} = \frac{0.66(7.17 + \frac{17746.56}{x_1 * x_5}) + \frac{17746.65}{x_1^2 x_5}(0.66(x_1 + x_2 - x_3 + 334))}{7.17 + \frac{17746.56}{x1*x5}^2} > 0$, $\frac{\partial y}{\partial x_2} = \frac{0.66}{7.17 + \frac{17746.56}{x_1 * x_5}} > 0$, $\frac{\partial y}{\partial x_3} = \frac{-0.66}{7.17 + \frac{17746.56}{x_1 x_5}} < 0$, $\frac{\partial y}{\partial x_4} = 0.66 > 0$, $\frac{\partial y}{\partial x_5} = \frac{0.66(x_1 + x_2 - x_3 + 334)}{7.17 + \frac{17746.56}{x_1 x_5}^2} \frac{17746.56}{x_1^2 x_5} > 0$ Based on the information, we can conclude that $x_1, x_2, x_4, x_5$ is directly proportional to $y$, and $x_3$ is inversely proportional to $y$. This is consistent with the correlation coefficient matrixE.1, which indicates that the expression obtained by our algorithm is an analyzable and reliable expression. Figure 3c displays the fitted curve of Equation 7 to the data.

## 5 DISSCUSION

We propose a new symbolic regression framework AlphaSymbol. The results are state-of-the-art on a series of benchmark symbolic regression tasks. In AlphaSymbol, we introduce an LSTM to guide MCTS in searching for good expressions. The data obtained from the MCTS simulation is then used to train the LSTM, which in turn provides better guidance for MCTS. To make the LSTM network more "confident", we introduce the information entropy loss of the predicted probability distribution $p$ in the loss function. Experimental results demonstrate that this operation can effectively improve the efficiency of the algorithm search. Secondly, in order to improve the problem of variable dropout that often occurs when the algorithm deals with multi-variable problems. We propose a new loss function, $S_{NRMSR}$, which not only considers $y$ and $\hat{y}$ but also considers $x_n$ using the Quasi-Euclidean distance between the real and predicted points in a multi-dimensional space as the loss. Our ablation experiments have demonstrated that introducing the proposed $S_{NRMSR}$ loss function yields a noteworthy enhancement in the algorithm's expression recovery rate, compared to using a regular loss function. Furthermore, this loss function effectively addresses the challenge of

variable omission that often arises when dealing with multi-variable problems.

When AlphaSymbol deals with data with too many variables, the search efficiency is low because of the large search space, and sometimes the target expression can not be obtained. In order to solve this problem, next we intend to pre-train the policy network with a large amount of data to improve the performance and efficiency of the algorithm.

Our future research will focus on the following aspect. In the physical world, real physical quantities have specific units, for instance, the unit of velocity is measured in meters per second (m/s). Hence, the expressions that we discover should be a variable measured in meters divided by a variable measured in seconds. With this prior knowledge, we can significantly enhance search efficiency. Therefore, in the following research, we will consider incorporating physical constraints into the search process based on the laws of the real physical world. This will result in a more scientifically sound and efficient symbol regression algorithm.

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

## A   APPENDIX: PSEUDOCODE FOR THE ALPHASYMBOL

**Algorithm 1** presents an overview of the AlphaSymbol framework. Prior to selecting a symbol, the $UseConstraint(\tau, s, \pi)$ function is applied to enforce constraints. The *counter* is updated after each symbol selection, and we check whether the *counter* is zero. If the *counter* is zero, the reward value of the obtained expression is computed and backpropagation is performed. If the current symbol is a leaf node but not a terminal node, we expand and evaluate the node.

---

**Algorithm 1:** AlphaSymbol

---

**Data:** $X = [x_1, x_2, ..., x_n]$ ;$y = [y_1, y_2, ..., y_n]$;S=$[+, -, \times, \div, ...]$
**Result:** Find an expression such that y = f(X)

1 initialization;
2 **while** *Reward* $\neq$ 1 **do**
3                                  ▷ A threshold can also be set, for example, Reward >= 0.9999.
4     **repeat**
5         ***Self-Search :***
6         UseConstraint($\tau, s, \pi$)3
7         symbol = arg max($\pi$)            ▷ Choosing the symbol with the highest probability
8         $\tau$.append(symbol)
9         $counter = counter$ + Arity(symbol)4 - 1    ▷ Whether or not the expression is complete
10         **if** *counter=0* **then**
11             z = $\frac{1}{1+S_{NRMSE}}$                       ▷ Calculating rewards
12             **if** $z > T$ **then**
13                         ▷ T represents the termination threshold of the algorithm
14                break;        ▷ Terminate the program upon achieving expected rewards
15             **end**
16             **Backpropagate :** $z$                    ▷ Backpropagate the final reward
17             Storing data : $[s, \pi, z]$
18             ***Train Neural Network :*** $\mathcal{N}_\theta \longrightarrow \mathcal{N}_{\theta NEW}$      ▷ Further training of LSTM
19         **end**
20         ***MCTS:***
21         **Expand and evaluate:**
22         $parent||sibling$ = ParentSibling($\tau_t$) 2           ▷ Get the neural network input
23         (p,v) =$\mathcal{N}_\theta(parent||sibling)$ ▷ Calculating probability distribution p and evaluat value v with LSTM
24         **for** $j \leftarrow 2$ **to** $n_{evaluate}$ **do**
25             **if** *current node = leafnode* & *counter* $\neq 0$ **then**
26                ***Expend(p)***                  ▷ *Expanding leaf nodes with probability p*
27             **else**
28                ***Select:*** $a_{t+1}$ = *arg max*($\mathcal{UCT}(s_t, a_t)$)     ▷ *Selecting the symbol with the largest UCT value as the next symbol*
29             **end**
30             ***Backpropagate(v)***                 ▷ *Backpropagate the evaluate value v*
31         **end**
32     **until** *Find the target epxression*;
33 **end**

---

**Algorithm 2** describes the function $ParentSibling(\tau)$ used in Algorithm 1 to find the parent and sibling nodes of the next symbol to the sample. This algorithm uses the following logic: If the last symbol in the partial traversal is a unary or binary operator and has no siblings, it is the parent node and its sibling is set to empty. Otherwise, the algorithm iterates forward in the traversal until it finds a

node with unselected children. This node is the parent and the subsequent node is the sibling.

---

**Algorithm 2:** ParentSibling($\tau$) (To retrieve the father and sibling nodes as inputs for an LSTM )

---

1 **Input :** Partially sampled traversal $\tau$
2 **Output :** Concatenated parent and sibling nodes of the next nodes to be generated
3 $T \leftarrow len(\tau)$                 ▷ Length of partial traversal
4 $counter \leftarrow 0$       ▷ Initializes a counter with no selected number of nodes
5 **if** $Arity(\tau_T) > 0$ **then**
6     $parent \leftarrow \tau_T$
7     $sibling \leftarrow empty$
8 **end**
9 **for** $i \leftarrow T$ **to** $1$ **do**
10     $counter = counter + Arity(\tau_i) - 1$
11                       ▷ Update counter of unselected nodes
12     **if** *counter = 0* **then**
13        $parent \leftarrow \tau_i$
14        $sibling \leftarrow \tau_{i+1}$
15     **end**
16 **end**

---

**Algorithm 3** demonstrates the application of a series of constraints during the symbol generation process. The specific steps are as follows: we first obtain the types of symbols in our symbol library, and then based on the current state, we sequentially determine whether each function in the symbol library should be "restricted". If a symbol is restricted, we set the probability of selecting that symbol to zero and finally normalize the probabilities of all symbols.

---

**Algorithm 3:** UseConstraints($\tau, S_i, \pi$)

---

1 **Input :** The simulated probability $\pi$; partially sampled traversal $\tau$; Used symbol library $S$
2 **Output :** The probability distribution $\pi$ adjusted according to the constraints
3 $L \leftarrow len(S)$                    ▷ Length of $S$
4 **for** $i \leftarrow 1$ **to** $L$ **do**
5     **if** $Constraint(\tau, S_i)$ **then**
6        $\pi_i \leftarrow 0$           ▷ Sets the restricted symbol probability to zero
7     **end**
8     $\pi \leftarrow \frac{\pi}{\sum_{i=0}^{L} \pi_i}$          ▷ The probability is normalized
9 **end**

---

**Algorithm 4** describes the Arity(s) function used in Algorithm 1, which obtains the arity of an operator. Specifically, for a variable $[x_1]$ and a constant $[c]$, Arity(s)=0. For a unary operator $[sin, cos, exp, ln, sqrt]$, Arity(s)=1. Similarly, for a binary operator, Arity(s)=2, and so on for other operators.

---

**Algorithm 4:** Arity($s$)

---

1 **Input :** Newly selected operator symbol $s$;
2 **Output :** The arity of an operator
3 $s \leftarrow select(S)$                  ▷ Selecting new symbols.
4 **if** $s \in [x_1, x_2, ..., x_n, c]$ **then**
5     return 0          ▷ If the symbol is a variable or a constant, the arity would be 0
6 **end**
7 **if** $s \in [sin, cos, exp, log, sqrt]$ **then**
8     return 1          ▷ If the operator is unary, the arity would be 1.
9 **end**
10 **if** $s \in [+, -, *, /]$ **then**
11     return 2          ▷ If the operator is a binary operator, it returns 2.
12 **end**

---

# B    APPENDIX: REWARD VARIATION CURVE (ON DATASET NGUYEN).

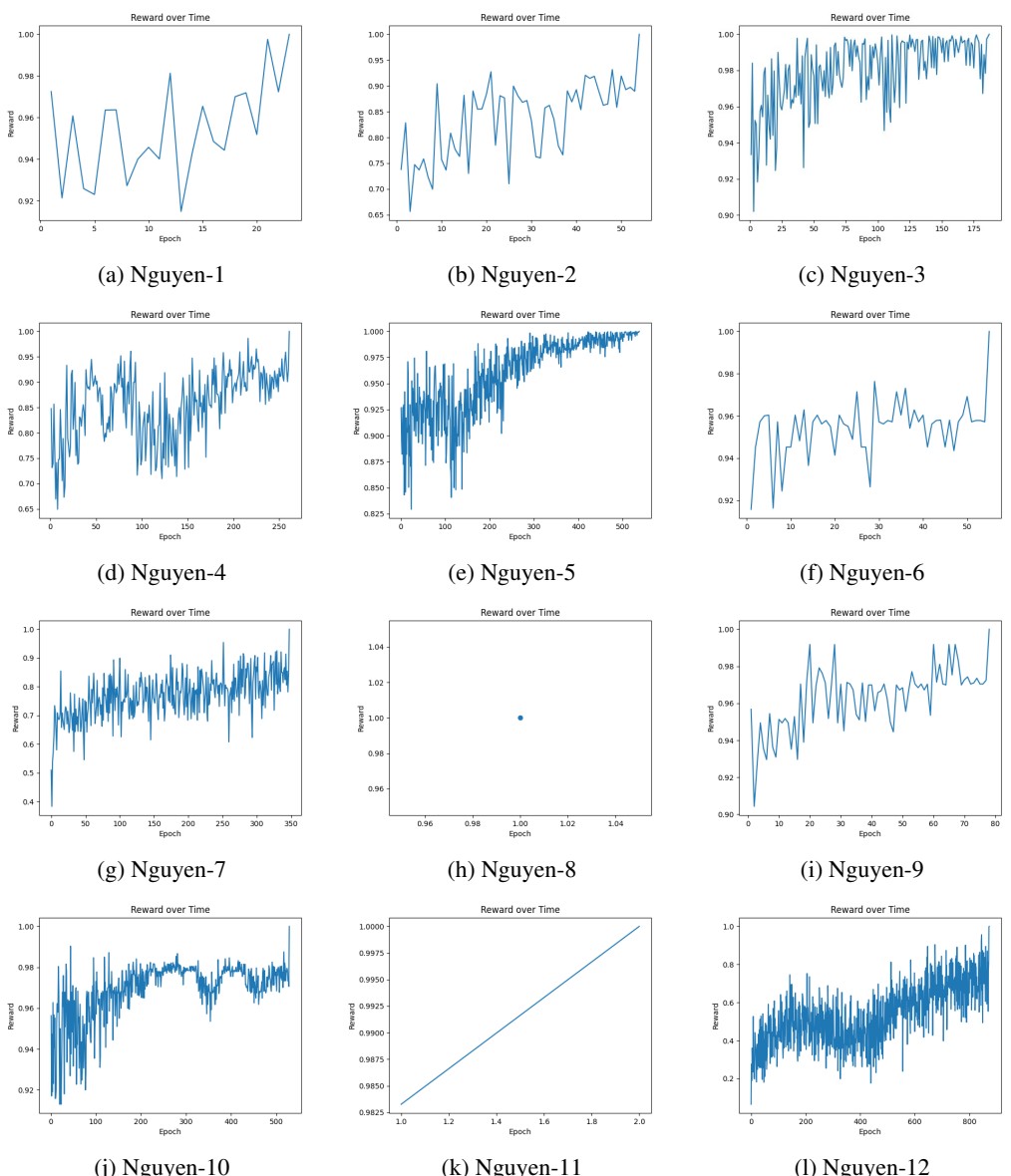

Figure B.1: The series of figures above presents line graphs depicting the reward values of AlphaSymbol on the Nguyen dataset over time. As observed from the figures, throughout the search process, the reward values for all expressions demonstrate an oscillatory ascent with the increase in training iterations. Notably, Expression 8 is an exception due to its comparatively simple structure, achieving its best result within just one epoch.

## C  APPENDIX: EXPERIMENTAL DETAILS FOR EACH EXPRESSION

Table $B.1 - B.3$ shows some specific details of different expressions when tested. The benchmark problem specifications for symbolic regression are as follows:

- Input variables are represented as $[x_1, x_2, ..., x_n]$

- $U(a, b, c)$ signifies $c$ random points uniformly sampled between $a$ and $b$ for each input variable. Different random seeds are used for training and testing datasets.

- $E(a, b, c)$ indicates $c$ points evenly spaced between $a$ and $b$ for each input variable.

- To simplify the notation, libraries are defined relative to a base library $[+, -, \times, \div, sin, cos, ln, exp, sqrt, x_1]$.

- Any unassigned operand is represented by $\bullet$, for instance, $\bullet^2$ corresponds to the square operator.

Table C.1: Symbol library and value range of the three data sets Nguyen, Korns, and Jin.

| Name | Expression | Dataset | Library |
|---|---|---|---|
| Nguyen-1 | $x_1^3 + x_1^2 + x_1$ | U(−1, 1, 20) | $\mathcal{L}_0$ |
| Nguyen-2 | $x_1^4 + x_1^3 + x_1^2 + x_1$ | U(−1, 1, 20) | $\mathcal{L}_0$ |
| Nguyen-3 | $x_1^5 + x_1^4 + x_1^3 + x_1^2 + x_1$ | U(−1, 1, 20) | $\mathcal{L}_0$ |
| Nguyen-4 | $x_1^6 + x_1^5 + x_1^4 + x_1^3 + x_1^2 + x_1$ | U(−1, 1, 20) | $\mathcal{L}_0$ |
| Nguyen-5 | $\sin(x_1^2)\cos(x) - 1$ | U(−1, 1, 20) | $\mathcal{L}_0$ |
| Nguyen-6 | $\sin(x_1) + \sin(x_1 + x_1^2)$ | U(−1, 1, 20) | $\mathcal{L}_0$ |
| Nguyen-7 | $\log(x_1 + 1) + \log(x_1^2 + 1)$ | U(0, 2, 20) | $\mathcal{L}_0$ |
| Nguyen-8 | $\sqrt{x}$ | U(0, 4, 20) | $\mathcal{L}_0$ |
| Nguyen-9 | $\sin(x) + \sin(x_2^2)$ | U(0, 1, 20) | $\mathcal{L}_0 \cup \{x_2\}$ |
| Nguyen-10 | $2\sin(x)\cos(x_2)$ | U(0, 1, 20) | $\mathcal{L}_0 \cup \{x_2\}$ |
| Nguyen-11 | $x_1^{x_2}$ | U(0, 1, 20) | $\mathcal{L}_0 \cup \{x_2, \bullet^\bullet, \text{const}\}$ |
| Nguyen-12 | $x_1^4 - x_1^3 + \frac{1}{2}x_2^2 - x_2$ | U(0, 1, 20) | $\mathcal{L}_0 \cup \{\bullet^4, \bullet^3, x_2, \text{const}\}$ |
| Nguyen-2′ | $4x_1^4 + 3x_1^3 + 2x_1^2 + x$ | U(−1, 1, 20) | $\mathcal{L}_0$ |
| Nguyen-5′ | $\sin(x_1^2)\cos(x) - 2$ | U(−1, 1, 20) | $\mathcal{L}_0$ |
| Nguyen-8′ | $\sqrt[3]{x}$ | U(0, 4, 20) | $\mathcal{L}_0 \cup \{\bullet^\bullet, \text{const}\}$ |
| Nguyen-8″ | $\sqrt[3]{x_1^2}$ | U(0, 4, 20) | $\mathcal{L}_0 \cup \{\bullet^\bullet, \text{const}\}$ |
| Nguyen-1$^c$ | $3.39x_1^3 + 2.12x_1^2 + 1.78x$ | U(−1, 1, 20) | $\mathcal{L}_0 \cup \{\text{const}\}$ |
| Nguyen-5$^c$ | $\sin(x_1^2)\cos(x) - 0.75$ | U(−1, 1, 20) | $\mathcal{L}_0 \cup \{\text{const}\}$ |
| Nguyen-7$^c$ | $\log(x + 1.4) + \log(x_1^2 + 1.3)$ | U(0, 2, 20) | $\mathcal{L}_0 \cup \{\text{const}\}$ |
| Nguyen-8$^c$ | $\sqrt{1.23x}$ | U(0, 4, 20) | $\mathcal{L}_0 \cup \{\text{const}\}$ |
| Nguyen-10$^c$ | $\sin(1.5x)\cos(0.5x_2)$ | U(0, 1, 20) | $\mathcal{L}_0 \cup \{x_2, \text{const}\}$ |
| Korns-1 | $1.57 + 24.3 * x_1^4$ | U(−1, 1, 20) | $\mathcal{L}_0$ |
| Korns-2 | $0.23 + 14.2\frac{(x_4 + x_1)}{(3x_2)}$ | U(−1, 1, 20) | $\mathcal{L}_0 \cup \{x_2, \text{const}\}$ |
| Korns-3 | $4.9\frac{(x_2 - x_1 + \frac{x_1}{x_3})}{(3x_3))} - 5.41$ | U(−1, 1, 20) | $\mathcal{L}_0 \cup \{x_2, x_3\text{const}\}$ |
| Korns-4 | $0.13sin(x_1) - 2.3$ | U(−1, 1, 20) | $\mathcal{L}_0$ |
| Korns-5 | $3 + 2.13log(|x_5|)$ | U(−1, 1, 20) | $\mathcal{L}_0 \cup \{\text{const}\}$ |
| Korns-6 | $1.3 + 0.13\sqrt{|x_1|}$ | U(−1, 1, 20) | $\mathcal{L}_0 \cup \{\text{const}\}$ |
| Korns-7 | $2.1(1 - e^{-0.55x_1})$ | U(−1, 1, 20) | $\mathcal{L}_0 \cup \{\text{const}\}$ |
| Korns-8 | $6.87 + 11\sqrt{|7.23x_1x_4x_5|}$ | U(−1, 1, 20) | $\mathcal{L}_0 \cup \{\text{const}\}$ |
| Korns-9 | $12\sqrt{|4.2x_1x_2x_2|}$ | U(−1, 1, 20) | $\mathcal{L}_0 \cup \{x_2, \text{const}\}$ |
| Korns-10 | $0.81 + 24.3\frac{2x_1 + 3x_2^2}{4x_3^3 + 5x_4^4}$ | U(−1, 1, 20) | $\mathcal{L}_0 \cup \{x_2, x_3, x_4, \text{const}\}$ |
| Korns-11 | $6.87 + 11cos(7.23x_1^3)$ | U(−1, 1, 20) | $\mathcal{L}_0 \cup \{x_2, \text{const}\}$ |
| Korns-12 | $2 - 2.1cos(9.8x_1^3)sin(1.3x_5)$ | U(−1, 1, 20) | $\mathcal{L}_0 \cup \{x_2, \text{const}\}$ |
| Korns-13 | $32.0 - 3.0\frac{tan(x_1)}{tan(x_2)}\frac{tan(x_3)}{tan(x_4)}$ | U(−1, 1, 20) | $\mathcal{L}_0 \cup \{x_2, x_3, x_4, \text{const}, tan, tanh\}$ |
| Korns-14 | $22.0 - (4.2cos(x_1) - tan(x_2))\frac{tanh(x_3)}{sin(x_4)}$ | U(−1, 1, 20) | $\mathcal{L}_0 \cup \{x_2, x_3, x_4, \text{const}, tan, tanh\}$ |
| Korns-15 | $12.0 - \frac{6.0tan(x_1)}{e^{x_2}}(log(x_3) - tan(x_4)))$ | U(−1, 1, 20) | $\mathcal{L}_0 \cup \{x_2, x_3, x_4, \text{const}, tan\}$ |
| Jin-1 | $2.5x_1^4 - 1.3x_1^3 + 0.5x_2^2 - 1.7x_2$ | U(−3, 3, 100) | $\mathcal{L}_0 - \{\log\} \cup \{\bullet^3, \bullet^4, x_2, \text{const}\}$ |
| Jin-2 | $8.0x_1^2 + 8.0x_2^3 - 15.0$ | U(−3, 3, 100) | $\mathcal{L}_0 - \{\log\} \cup \{\bullet^2, \bullet^3, x_2,$ |
| Jin-3 | $0.2x_1^3 + 0.5x_2^3 - 1.2x_2 - 0.5x_1$ | U(−3, 3, 100) | $\mathcal{L}_0 - \{\log\} \cup \{\bullet^3, x_2, \text{const}\}$ |
| Jin-4 | $1.5\exp x + 5.0cos(x_2)$ | U(−3, 3, 100) | $\mathcal{L}_0 - \{\log\} \cup \{\bullet^2, \bullet^3, x_2,$ |
| Jin-5 | $6.0sin(x_1)cos(x_2)$ | U(−3, 3, 100) | $\mathcal{L}_0 - \{\log\} \cup \{\bullet^2, \bullet^3, x_2,$ |
| Jin-6 | $1.35x_1x_2 + 5.5sin((x_1 - 1.0)(x_2 - 1.0))$ | U(−3, 3, 100) | $\mathcal{L}_0 - \{\log\} \cup \{\bullet^2, \bullet^3, x_2,$ |

Table C.2: Symbol library and value range of the three data sets neat, Keijzer and Livermore.

| Name | Expression | Dataset | Library |
|---|---|---|---|
| Neat-1 | $x_1^4 + x_1^3 + x_1^2 + x$ | U$(-1, 1, 20)$ | $\mathcal{L}_0$ |
| Neat-2 | $x_1^5 + x_1^4 + x_1^3 + x_1^2 + x$ | U$(-1, 1, 20)$ | $\mathcal{L}_0$ |
| Neat-3 | $\sin(x_1^2)\cos(x) - 1$ | U$(-1, 1, 20)$ | $\mathcal{L}_0 \cup \{1\}$ |
| Neat-4 | $\log(x + 1) + \log(x_1^2 + 1)$ | U$(0, 2, 20)$ | $\mathcal{L}_0 \cup \{1\}$ |
| Neat-5 | $2\sin(x)\cos(x_2)$ | U$(-1, 1, 100)$ | $\mathcal{L}_0 \cup \{x_2\}$ |
| Neat-6 | $\sum_{k=1}^{x} \frac{1}{k}$ | E$(1, 50, 50)$ | $\{+, \times, \div, \bullet^{-1}, -\bullet, \sqrt{\bullet}, x\}$ |
| Neat-7 | $2 - 2.1\cos(9.8x_1)\sin(1.3x_2)$ | E$(-50, 50, 10^5)$ | $\mathcal{L}_0 \cup \{x_2\}$ |
| Neat-8 | $\frac{e^{-(x_1)^2}}{1.2+(x_2-2.5)^2}$ | U$(0.3, 4, 100)$ | $\{+, -, \times, \div, \exp, e^{-\bullet}, \bullet^2, x, x_2\}$ |
| Neat-9 | $\frac{1}{1+x_1^{-4}} + \frac{1}{1+x_2^{-4}}$ | E$(-5, 5, 21)$ | $\mathcal{L}_0 \cup \{x_2\}$ |
| Keijzer-1 | $0.3x_1 sin(2\pi x_1)$ | U$(-1, 1, 20)$ | $\mathcal{L}_0 \cup \{const\}$ |
| Keijzer-2 | $2.0x_1 sin(0.5\pi x_1)$ | U$(-1, 1, 20)$ | $\mathcal{L}_0 \cup \{const\}$ |
| Keijzer-3 | $0.92x_1 sin(2.41\pi x_1)$ | U$(-1, 1, 20)$ | $\mathcal{L}_0 \cup \{const\}$ |
| Keijzer-4 | $x_1^3 e^{-x_1} cos(x_1) sin(x_1) sin(x_1)^2 cos(x_1) - 1$ | U$(-1, 1, 20)$ | $\mathcal{L}_0 \cup \{1\}$ |
| Keijzer-5 | $3 + 2.13 log(|x_5|)$ | U$(-1, 1, 20)$ | $\mathcal{L}_0 \cup \{const\}$ |
| Keijzer-6 | $\frac{x1(x1+1)}{2}$ | U$(-1, 1, 20)$ | $\mathcal{L}_0 \cup \{const\}$ |
| Keijzer-7 | $log(x_1)$ | U$(0, 1, 20)$ | $\mathcal{L}_0$ |
| Keijzer-8 | $\sqrt{(x_1)}$ | U$(0, 1, 20)$ | $\mathcal{L}_0$ |
| Keijzer-9 | $log(x_1 + \sqrt{x_1^2 + 1})$ | U$(-1, 1, 20)$ | $\mathcal{L}_0 \cup \{x_2\}$ |
| Keijzer-10 | $x_1^{x_2}$ | U$(-1, 1, 20)$ | $\mathcal{L}_0 \cup \{x_2, \bullet^\bullet, const\}$ |
| Keijzer-11 | $x_1 x_2 + sin((x_1 - 1)(x_2 - 1))$ | U$(-1, 1, 20)$ | $\mathcal{L}_0 \cup \{x_2, 1\}$ |
| Keijzer-12 | $x_1^4 - x_1^3 + \frac{x_2^2}{2} - x_2$ | U$(-1, 1, 20)$ | $\mathcal{L}_0 - \{\log\} \cup \{\bullet^3, \bullet^4, x_2, const\}$ |
| Keijzer-13 | $6sin(x_1)cos(x_2)$ | U$(-1, 1, 20)$ | $\mathcal{L}_0 \cup \{x_2, const\}$ |
| Keijzer-14 | $\frac{8}{2+x_1^2+x_2^2}$ | U$(-1, 1, 20)$ | $\mathcal{L}_0 \cup \{x_2, const\}$ |
| Keijzer-15 | $\frac{x_1^3}{5} + \frac{x_2^3}{2} - x_2 - x_1$ | U$(-1, 1, 20)$ | $\mathcal{L}_0 - \{\log\} \cup \{\bullet^3, x_2, const\}$ |
| Livermore-1 | $\frac{1}{3} + x_1 + sin(x_1^2))$ | U$(-3, 3, 100)$ | $\mathcal{L}_0 - \{\log\}$ |
| Livermore-2 | $sin(x_1^2) * cos(x1) - 2$ | U$(-3, 3, 100)$ | $\mathcal{L}_0 - \{\log\}$ |
| Livermore-3 | $sin(x_1^3) * cos(x_1^2)) - 1$ | U$(-3, 3, 100)$ | $\mathcal{L}_0 - \{\log\}$ |
| Livermore-4 | $log(x_1 + 1) + log(x_1^2 + 1) + log(x_1)$ | U$(-3, 3, 100)$ | $\mathcal{L}_0 \cup \{1\}$ |
| Livermore-5 | $x_1^4 - x_1^3 + x_2^2 - x_2$ | U$(-3, 3, 100)$ | $\mathcal{L}_0 - \{\log\} \cup \{\bullet^3, \bullet^4, x_2, const\}$ |
| Livermore-6 | $4x_1^4 + 3x_1^3 + 2x_1^2 + x_1$ | U$(-3, 3, 100)$ | $\mathcal{L}_0 - \{\log\} \cup \{\bullet^3, \bullet^4, x_2, const\}$ |
| Livermore-7 | $\frac{(exp(x1)-exp(-x_1)}{2})$ | U$(-1, 1, 100)$ | $\mathcal{L}_0 \cup \{const\}$ |
| Livermore-8 | $\frac{(exp(x1)+exp(-x1)}{3}$ | U$(-3, 3, 100)$ | $\mathcal{L}_0 \cup \{const\}$ |
| Livermore-9 | $x_1^9 + x_1^8 + x_1^7 + x_1^6 + x_1^5 + x_1^4 + x_1^3 + x_1^2 + x_1$ | U$(-1, 1, 100)$ | $\mathcal{L}_0$ |
| Livermore-10 | $6 * sin(x_1)cos(x_2)$ | U$(-3, 3, 100)$ | $\mathcal{L}_0 \cup \{x_2, const\}$ |
| Livermore-11 | $\frac{x_1^2 x_2^2}{(x_1+x_2)}$ | U$(-3, 3, 100)$ | $\mathcal{L}_0 \cup \{x_2\}$ |
| Livermore-12 | $\frac{x_1^5}{x_2^3}$ | U$(-3, 3, 100)$ | $\mathcal{L}_0 \cup \{x_2\}$ |
| Livermore-13 | $x_1^{\frac{1}{3}}$ | U$(-3, 3, 100)$ | $\mathcal{L}_0 \cup \{\bullet^\bullet, const\}$ |
| Livermore-14 | $x_1^3 + x_1^2 + x_1 + sin(x_1) + sin(x_2^2)$ | U$(-1, 1, 100)$ | $\mathcal{L}_0 - \{\log\} \cup \{\bullet^3, \bullet^4, x_2, const\}$ |
| Livermore-15 | $x_1^{\frac{1}{5}}$ | U$(-3, 3, 100)$ | $\mathcal{L}_0 \cup \{\bullet^\bullet, const\}$ |
| Livermore-16 | $x_1^{\frac{2}{3}}$ | U$(-3, 3, 100)$ | $\mathcal{L}_0 \cup \{\bullet^\bullet, const\}$ |
| Livermore-17 | $4sin(x_1)cos(x_2)$ | U$(-3, 3, 100)$ | $\mathcal{L}_0 \cup \{x_2, const\}$ |
| Livermore-18 | $sin(x_1^2) * cos(x_1) - 5$ | U$(-3, 3, 100)$ | $\mathcal{L}_0 \cup \{const\}$ |
| Livermore-19 | $x_1^5 + x_1^4 + x_1^2 + x_1$ | U$(-3, 3, 100)$ | $\mathcal{L}_0$ |
| Livermore-20 | $e^{(-x_1^2)}$ | U$(-3, 3, 100)$ | $\mathcal{L}_0 \cup \{-1\}$ |
| Livermore-21 | $x_1^8 + x_1^7 + x_1^6 + x_1^5 + x_1^4 + x_1^3 + x_1^2 + x_1$ | U$(-1, 1, 20)$ | $\mathcal{L}_0$ |
| Livermore-22 | $e^{(-0.5x_1^2)}$ | U$(-3, 3, 100)$ | $\mathcal{L}_0 \cup \{const\}$ |

Table C.3: Symbol library and value range of the three data sets Vladislavleva and others.

| Name | Expression | Dataset | Library |
|---|---|---|---|
| Vladislavleva-1 | $\frac{(e^{-(x_1-1)^2})}{(1.2+(x_2-2.5)^2))}$ | $U(-1,1,20)$ | $\mathcal{L}_0 \cup \{const\}$ |
| Vladislavleva-2 | $e^{-x_1}x_1^3cos(x_1)sin(x_1)(cos(x_1)sin(x_1)^2-1)$ | $U(-1,1,20)$ | $\mathcal{L}_0 - \{log\} \cup \{\bullet^2, \bullet^3, x_2,$ |
| Vladislavleva-3 | $e^{-x_1}x_1^3cos(x_1)sin(x_1)(cos(x_1)sin(x_1)^2-1)(x_2-5)$ | $U(-1,1,20)$ | $\mathcal{L}_0 \cup \{1\}$ |
| Vladislavleva-4 | $\frac{10}{5+(x_1-3)^2+(x_2-3)^2+(x_3-3)^2+(x_4-3)^2+(x_5-3)^2}$ | $U(0,2,20)$ | $\mathcal{L}_0 \cup \{x_2, x_3, x_4, x_5, \bullet^\bullet, const, tan\}$ |
| Vladislavleva-5 | $30(x_1-1)\frac{x_3-1}{(x_1-10)}x_2^2$ | $U(-1,1,100)$ | $\mathcal{L}_0 \cup \{x_2, const\}$ |
| Vladislavleva-6 | $6sin(x_1)cos(x_2)$ | $E(1,50,50)$ | $\mathcal{L}_0 \cup \{x_2, const\}$ |
| Vladislavleva-7 | $2-2.1\cos(9.8x)\sin(1.3x_2)$ | $E(-50,50,10^5)$ | $\mathcal{L}_0 \cup \{x_2, const\}$ |
| Vladislavleva-8 | $\frac{e^{-(x-1)^2}}{1.2+(x_2-2.5)^2}$ | $U(0.3,4,100)$ | $\{+,-,\times,\div, exp, e^{-\bullet}, \bullet^2, x, x_2\}$ |
| Test-2 | $3.14*x1*x1$ | $U(-1,1,20)$ | $\mathcal{L}_0 \cup \{const\}$ |
| Const-Test-1 | $5*x1*x1$ | $U(-1,1,20)$ | $\mathcal{L}_0 \cup \{const\}$ |
| GrammarVAE-1 | $1./3 + x1 + sin(x_1^2))$ | $U(-1,1,20)$ | $\mathcal{L}_0 \cup \{const\}$ |
| Sine | $sin(x_1) + sin(x_1 + x_1^2))$ | $U(-1,1,20)$ | $\mathcal{L}_0$ |
| Nonic | $x_1^9+x_1^8+x_1^7+x_1^6+x_1^5+x_1^4+x_1^3+x_1^2+x_1$ | $U(-1,1,100)$ | $\mathcal{L}_0 \cup \{x_2\}$ |
| Pagie-1 | $\frac{1}{1+x_1^{-4}}+\frac{1}{1+x2^{-4}}$ | $E(1,50,50)$ | $\mathcal{L}_0 \cup \{x_2\}$ |
| Meier-3 | $\frac{x_1^2x_2^2}{(x_1+x_2)}$ | $E(-50,50,10^5)$ | $\mathcal{L}_0 \cup \{x_2\}$ |
| Meier-4 | $\frac{x_1^5}{x_2^3}$ | $U(0.3,4,100)$ | $\{+,-,\times,\div, exp, e^{-\bullet}, \bullet^2, x, x_2\}$ |
| Poly-10 | $x_1x_2 + x_3x4 + x_5x_6 + x_1x_7x_9 + x_3x_6x_{10}$ | $E(-1,1,100)$ | $\mathcal{L}_0 \cup \{x_2, x_3, x_4, x_5, x_6, x_7, x_8, x_9, x_{10}\}$ |

# D   Appendix: Average Coefficient of Determination ($R^2$) on Various Datasets

To assess the goodness of fit of AlphaSymbol on the datasets, we also recorded the average R2 of AlphaSymbol on various testing datasets. From the table, it can be observed that AlphaSymbol achieved an average R2 exceeding 0.99 on the majority of the datasets. This suggests that while AlphaSymbol may not be able to fully recover the original formula of certain expressions, it can still find an equivalent expression that fits the observed data well.

Table D.1: Average Coefficient of Determination ($R^2$) on Various Datasets

| Benchmark | $R^2$ |
|---|---|
| Nguyen | 0.9999 |
| Keijzer | 0.9991 |
| Korns | 0.9982 |
| Constant | 0.9991 |
| Livermore | 0.9998 |
| Vladislavlev | 0.9831 |
| R | 0.9702 |
| Jin | 0.9888 |
| Neat | 0.9763 |
| AI Feynman | 0.9960 |
| Others | 0.9982 |
| **Average** | **0.9917** |

# E   Appendix: $R^2$ of AlphaSymbol on the AI Feynman dataset.

We tested the performance of our proposed symbol regression algorithm, AlphaSymbol, on the AI Feynman dataset. This dataset contains problems from physics and mathematics across multiple subfields, such as mechanics, thermodynamics, and electromagnetism. The authors provided 100,000 sampled data points in the AI Feynman dataset, however, to better test the performance of AlphaSymbol, we randomly selected only 100 data points from the 100,000 provided as our experimental data. We applied AlphaSymbol to perform symbol regression on each data in the dataset. and recorded the $R^2$ between the predicted results and the correct answers. The experimental results indicate that

AlphaSymbol can accurately fit the corresponding expressions from a small number of sample points. For the majority of the formulas, the $R^2$ exceeds 0.99. This indicates that the model performs well on problems in fields such as physics and mathematics, and has great potential for wide application. The experimental results are shown in Table E.1 and Table E.2.

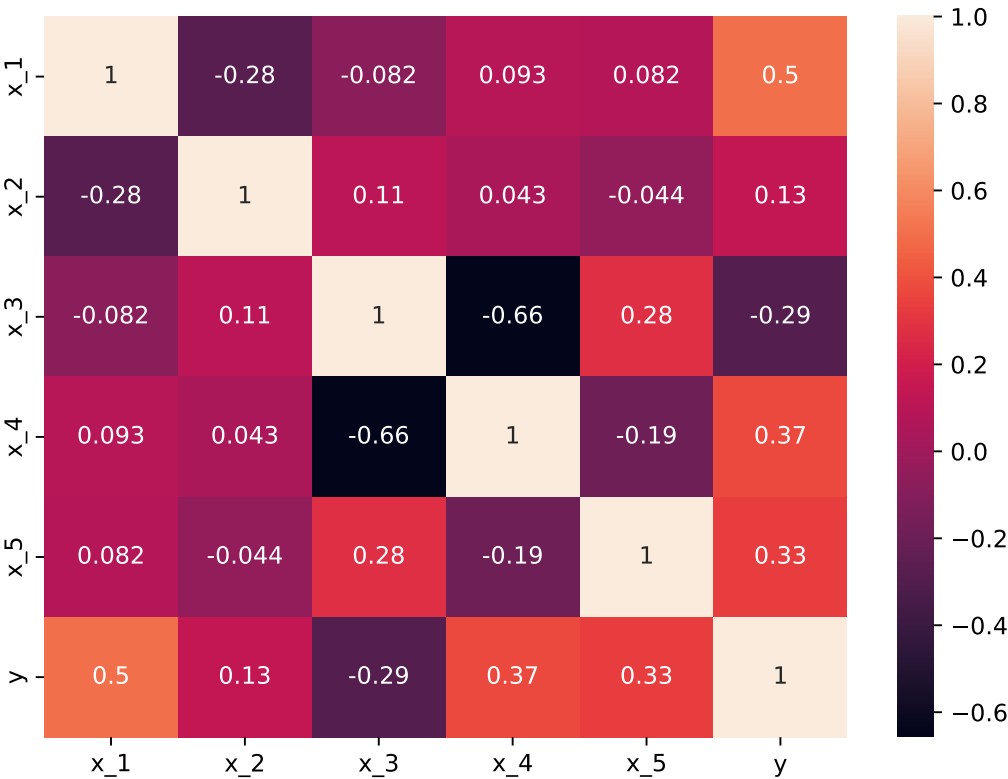

Figure E.1: This is a heatmap of the correlation coefficients between each variable $x$ and $y$. From the image, we can see that y has a positive correlation with $x_1$, $x_2$, $x_4$, and $x_5$ to varying degrees, and has a negative correlation with $x_3$.

## F  APPENDIX: HYPERPARAMETER SETTINGS

- **min_length (int):** minimum number of operators to allow in expression
- **max_length (int):** maximum number of operators to allow in expression
- **type ('rnn', 'lstm', or 'gru'):** type of architecture to use
- **num_layers (int):** number of layers in RNN architecture
- **dropout (float):** dropout (if any) for RNN architecture
- **lr (float):** learning rate for RNN
- **optimizer ('adam' or 'rmsprop'):** optimizer for RNN
- **inner_optimizer ('lbfgs', 'adam', or 'rmsprop'):** optimizer for expressions
- **inner_lr (float):** learning rate for constant optimization
- **inner_num_epochs (int):** number of epochs for constant optimization
- **batch_size (int):** batch size for training the RNN
- $\lambda$ **(float):** The weight assigned to the X part of the loss, ranges from 0 to 1.

when training the RNN.
- **batch_size (int):** batch size for training the RNN
- **num_batches (int):** number of batches (will stop early if found)
- **hidden_size (int):** hidden dimension size for RNN

Table E.1: Tested Feynman Equations, part 1.

| Feynman | Equation | $R^2$ |
|---|---|---|
| I.6.20a | $f = e^{-\theta^2/2}/\sqrt{2\pi}$ | 0.9992 |
| I.6.20 | $f = e^{-\frac{\theta^2}{2\sigma^2}}/\sqrt{2\pi\sigma^2}$ | 0.9988 |
| I.6.20b | $f = e^{-\frac{(\theta-\theta_1)^2}{2\sigma^2}}/\sqrt{2\pi\sigma^2}$ | 0.9923 |
| I.8.14 | $d = \sqrt{(x_2-x_1)^2+(y_2-y_1)^2}$ | 0.8929 |
| I.9.18 | $F = \frac{Gm_1m_2}{(x_2-x_1)^2+(y_2-y_1)^2+(z_2-z_1)^2}$ | 0.9944 |
| I.10.7 | $F = \frac{Gm_1m_2}{(x_2-x_1)^2+(y_2-y_1)^2+(z_2-z_1)^2}$ | 0.9906 |
| I.11.19 | $A = x_1y_1 + x_2y_2 + x_3y_3$ | 1.0 |
| I.12.1 | $F = \mu N_n$ | 1.0 |
| I.12.2 | $F = \frac{q_1q_2}{4\pi\epsilon r^2}$ | 1.0 |
| I.12.4 | $E_f = \frac{q_1}{4\pi\epsilon r^2}$ | 0.9994 |
| I.12.5 | $F = q_2 E_f$ | 1.0 |
| I.12.11 | $F = Q(E_f + Bv\sin\theta)$ | 0.9999 |
| I.13.4 | $K = \frac{1}{2}m(v^2+u^2+w^2)$ | 0.9969 |
| I.13.12 | $U = Gm_1m_2(\frac{1}{r_2}-\frac{1}{r_1})$ | 1.0 |
| I.14.3 | $U = mgz$ | 1.0 |
| I.14.4 | $U = \frac{k_{spring}x^2}{2}$ | 0.9999 |
| I.15.3x | $x_1 = \frac{x-ut}{\sqrt{1-u^2/c^2}}$ | 0.9993 |
| I.15.3t | $t_1 = \frac{t-ux/c^2}{\sqrt{1-u^2/c^2}}$ | 0.9844 |
| I.15.10 | $p = \frac{m_0v}{\sqrt{1-v^2/c^2}}$ | 0.9978 |
| I.16.6 | $v_1 = \frac{u+v}{1+uv/c^2}$ | 0.9873 |
| I.18.4 | $r = \frac{m_1r_1+m_2r_2}{m_1+m_2}$ | 0.9894 |
| I.18.12 | $\tau = rF\sin\theta$ | 0.9999 |
| I.18.16 | $L = mrv\sin\theta$ | 0.9999 |
| I.24.6 | $E = \frac{1}{4}m(\omega^2+\omega_0^2)x^2$ | 0.9986 |
| I.25.13 | $V_e = \frac{q}{C}$ | 1.0 |
| I.26.2 | $\theta_1 = \arcsin(n\sin\theta_2)$ | 0.9991 |
| I.27.6 | $f_f = \frac{1}{\frac{1}{d_1}+\frac{n}{d_2}}$ | 0.9995 |
| I.29.4 | $k = \frac{\omega}{c}$ | 1.0 |
| I.29.16 | $x = \sqrt{x_1^2+x_2^2-2x_1x_2\cos(\theta_1-\theta_2)}$ | 0.9942 |
| I.30.3 | $I_* = I_{*0}\frac{\sin^2(n\theta/2)}{\sin^2(\theta/2)}$ | 0.9912 |
| I.30.5 | $\theta = \arcsin(\frac{\lambda}{nd})$ | 0.9994 |
| I.32.5 | $P = \frac{q^2a^2}{6\pi\epsilon c^3}$ | 0.9857 |
| I.32.17 | $P = (\frac{1}{2}\epsilon cE_f^2)(8\pi r^2/3)(\omega^4/(\omega^2-\omega_0^2)^2)$ | 0.9788 |
| I.34.8 | $\omega = \frac{qvB}{p}$ | 1.0 |
| I.34.10 | $\omega = \frac{\omega_0}{1-v/c}$ | 0.9928 |
| I.34.14 | $\omega = \frac{1+v/c}{\sqrt{1-v^2/c^2}}\omega_0$ | 0.9992 |
| I.34.27 | $E = \hbar\omega$ | 1.0 |
| I.37.4 | $I_* = I_1 + I_2 + 2\sqrt{I_1I_2}\cos\delta$ | 0.9927 |
| I.38.12 | $r = \frac{4\pi\epsilon\hbar^2}{mq^2}$ | 0.9999 |
| I.39.10 | $E = \frac{3}{2}p_FV$ | 1.0 |
| I.39.11 | $E = \frac{1}{\gamma-1}p_FV$ | 0.9998 |
| I.39.22 | $P_F = \frac{nk_bT}{V}$ | 0.9999 |
| I.40.1 | $n = n_0e^{-\frac{mgx}{k_bT}}$ | 0.9947 |
| I.41.16 | $L_{rad} = \frac{\hbar\omega^3}{\pi^2c^2(e^{\frac{\hbar\omega}{k_bT}}-1)}$ | 0.8462 |
| I.43.16 | $v = \frac{\mu_{drift}qV_e}{d}$ | 1.0 |
| I.43.31 | $D = \mu_e k_bT$ | 1.0 |
| I.43.43 | $\kappa = \frac{1}{\gamma-1}\frac{k_bv}{A}$ | 0.9428 |
| I.44.4 | $E = nk_bT\ln(\frac{V_2}{V_1})$ | 0.8322 |
| I.47.23 | $c = \sqrt{\frac{\gamma pr}{\rho}}$ | 0.9926 |
| I.48.20 | $E = \frac{mc^2}{\sqrt{1-v^2/c^2}}$ | 0.8859 |
| I.50.26 | $x = x_1[\cos(\omega t) + \alpha\cos(\omega t)^2]$ | 0.9999 |

Table E.2: Tested Feynman Equations, part 2.

| Feynman | Equation | $R^2$ |
|---|---|---|
| II.2.42 | $P = \frac{\kappa(T_2 - T_1)A}{d}$ | 0.7842 |
| II.3.24 | $F_E = \frac{P}{4\pi r^2}$ | 0.9976 |
| II.4.23 | $V_e = \frac{q}{4\pi\epsilon r}$ | 0.9997 |
| II.6.11 | $V_e = \frac{1}{4\pi\epsilon}\frac{p_d\cos\theta}{r^2}$ | 1.0 |
| II.6.15a | $E_f = \frac{3}{4\pi\epsilon}\frac{p_d z}{r^5}\sqrt{x^2 + y^2}$ | 0.9466 |
| II.6.15b | $E_f = \frac{3}{4\pi\epsilon}\frac{p_d}{r^3}\cos\theta\sin\theta$ | 0.9943 |
| II.8.7 | $E = \frac{3}{5}\frac{q^2}{4\pi\epsilon d}$ | 0.9955 |
| II.8.31 | $E_{den} = \frac{\epsilon E_f^2}{2}$ | 1.0 |
| II.10.9 | $E_f = \frac{\sigma_{den}}{\epsilon}\frac{1}{1+\chi}$ | 0.9999 |
| II.11.3 | $x = \frac{qE_f}{m(\omega_0^2 - \omega^2)}$ | 0.9901 |
| II.11.7 | $n = n_0(1 + \frac{p_d E_f\cos\theta}{k_b T})$ | 0.8826 |
| II.11.20 | $P_* = \frac{n_\rho p_d^2 E_f}{3k_b T}$ | 0.7783 |
| II.11.27 | $P_* = \frac{n\alpha}{1 - n\alpha/3}\epsilon E_f$ | 0.9859 |
| II.11.28 | $\theta = 1 + \frac{n\alpha}{1 - (n\alpha/3)}$ | 0.9947 |
| II.13.17 | $B = \frac{1}{4\pi\epsilon c^2}\frac{2I}{r}$ | 0.9997 |
| II.13.23 | $\rho_c = \frac{\rho_{c_0}}{\sqrt{1 - v^2/c^2}}$ | 0.9807 |
| II.13.34 | $j = \frac{\rho_{c_0} v}{\sqrt{1 - v^2/c^2}}$ | 0.9938 |
| II.15.4 | $E = -\mu_M B\cos\theta$ | 1.0 |
| II.15.5 | $E = -p_d E_f\cos\theta$ | 1.0 |
| II.21.32 | $V_e = \frac{q}{4\pi\epsilon r(1 - v/c)}$ | 0.9954 |
| II.24.17 | $k = \sqrt{\frac{\omega^2}{c^2} - \frac{\pi^2}{d^2}}$ | 0.9872 |
| II.27.16 | $F_E = \epsilon c E_f^2$ | 1.0 |
| II.27.18 | $E_{den} = \epsilon E_f^2$ | 1.0 |
| II.34.2a | $I = \frac{qv}{2\pi r}$ | 0.9982 |
| II.34.2 | $\mu_M = \frac{qvr}{2}$ | 0.9918 |
| II.34.11 | $\omega = \frac{g\,qB}{2m}$ | 0.9937 |
| II.34.29a | $\mu_M = \frac{qh}{4\pi m}$ | 1.0 |
| II.34.29b | $E = \frac{g\,\mu_M B J_z}{\hbar}$ | 0.8882 |
| II.35.18 | $n = \frac{n_0}{\exp(\mu_m B/(k_b T)) + \exp(-\mu_m B/(k_b T))}$ | 0.9466 |
| II.35.21 | $M = n_\rho\mu_M\tanh(\frac{\mu_M B}{k_b T})$ | 0.8722 |
| II.36.38 | $f = \frac{\mu_m B}{k_b T} + \frac{\mu_m\alpha M}{\epsilon c^2 k_b T}$ | 0.9244 |
| II.37.1 | $E = \mu_M(1 + \chi)B$ | 0.9999 |
| II.38.3 | $F = \frac{YAx}{d}$ | 1.0 |
| II.38.14 | $\mu_S = \frac{Y}{2(1+\sigma)}$ | 0.9999 |
| III.4.32 | $n = \frac{1}{e^{\frac{\hbar\omega}{k_b T}} - 1}$ | 0.9877 |
| III.4.33 | $E = \frac{\hbar\omega}{e^{\frac{\hbar\omega}{k_b T}} - 1}$ | 0.9998 |
| III.7.38 | $\omega = \frac{2\mu_M B}{\hbar}$ | 0.9914 |
| III.8.54 | $p_\gamma = \sin(\frac{Et}{\hbar})^2$ | 0.9943 |
| III.9.52 | $p_\gamma = \frac{p_d E_f t}{\hbar}\frac{\sin((\omega - \omega_0)t/2)^2}{((\omega - \omega_0)t/2)^2}$ | 0.7266 |
| III.10.19 | $E = \mu_M\sqrt{B_x^2 + B_y^2 + B_z^2}$ | 0.9928 |
| III.12.43 | $L = n\hbar$ | 1.0 |
| III.13.18 | $v = \frac{2Ed^2 k}{\hbar}$ | 0.9999 |
| III.14.14 | $I = I_0(e^{\frac{qV_e}{k_b T}} - 1)$ | 0.9982 |
| III.15.12 | $E = 2U(1 - \cos(kd))$ | 0.9999 |
| III.15.14 | $m = \frac{\hbar^2}{2Ed^2}$ | 0.9983 |
| III.15.27 | $k = \frac{2\pi\alpha}{nd}$ | 0.9998 |
| III.17.37 | $f = \beta(1 + \alpha\cos\theta)$ | 1.0 |
| III.19.51 | $E = \frac{-mq^4}{2(4\pi\epsilon)^2\hbar^2}\frac{1}{n^2}$ | 0.9894 |
| III.21.20 | $j = \frac{-\rho_{c_0} q A_{vec}}{m}$ | 0.7489 |

**- use_gpu (bool):** whether or not to train with GPU

Table F.1: Tuned hyperparameters for AlphaSymbol.

| Parameter | Value |
|---|---|
| min_length | 2 |
| max_length | - |
| type | LSTM |
| Num_layers for LSTM | 2 |
| hidden_size | 250 |
| dropout | 0.0 |
| optimizer for LSTM | adam |
| Learn rate | 0.0005 |
| batch_size | 1000 |
| Use_gpu | False |
| inner_optimizer | lbfgs |
| inner_lr | 0.1 |
| inner_num_epochs | 5 |
| num_batches | 10000 |
| $\lambda$ | 0.1 |

## G  APPENDIX: RELATED WORK SUPPLEMENT

**Self-Learning_Gene_Expression_Programming (SL-GEP)**(Zhong et al., 2015), The SL-GEP method utilizes Gene Expression Programming (GEP) to represent each chromosome, which consists of a main program and a set of Automatically Defined Functions (ADFs). Each ADF is a sub-function used to solve sub-problems and is combined with the main program to address the target problem of interest. In the initialization step, all encoded ADFs in each chromosome are randomly generated. Then, during the evolutionary search process, SL-GEP employs a self-learning mechanism to improve the search outcomes. Specifically, SL-GEP utilizes an adaptive learning algorithm to dynamically evolve the ADFs online and integrate them with the main program to construct more complex and higher-order sub-functions, thereby enhancing search accuracy and efficiency.

**semantic genetic programming (SGD)**(Huang et al., 2022), Traditional genetic programming approaches often rely on random search to find optimal solutions, but this method is inefficient and prone to getting stuck in local optima. Therefore, SGD utilizes program behavior to guide the search, aiming to improve the efficiency and accuracy of symbolic regression problems. Specifically, this method starts by transforming input data into vector form and uses it as a constraint in a linear programming model. Then, semantic information is employed to evaluate each program and classify them based on their behavioral characteristics. Subsequently, the best programs are selected within each category and used to generate a new generation of programs. This approach effectively reduces the search space and accelerates convergence speed.

**shape-constrained symbolic regression (SCSR)** (Haider et al., 2023), The main idea of SCSR is a shape-constrained symbolic regression algorithm. This method leverages prior knowledge about the shape of the regression function to improve the accuracy of the regression model. Specifically, the article introduces both single-objective and multi-objective algorithms. The single-objective algorithm utilizes genetic programming techniques to generate the best-fitting curve. On the other hand, the multi-objective algorithm considers multiple optimization objectives and employs Pareto front techniques to search for a set of non-dominated solutions.

## H  APPENDIX: COMPUTING RESOURCES

The server we use is equipped with an Intel(R) Xeon(R) Gold 5218R CPU, which has a base frequency of 2.10 GHz. It has a total of 20 CPU cores, allowing for parallel processing and improved

computational performance. The high core count and efficient architecture of the Intel Xeon Gold 5218R make it suitable for handling demanding computational tasks and workloads.

# I   APPENDIX: MCTS

In order to clearly show the MCTS search process, we assume that there are only two basic symbols [sin,x]. The target expression is y = sin(x). The search process is as follows.

**Initialization:** Initially there is a root node $S_0$, and each node in the tree has two values, the reward Q of the node and the number of visits to that node N.

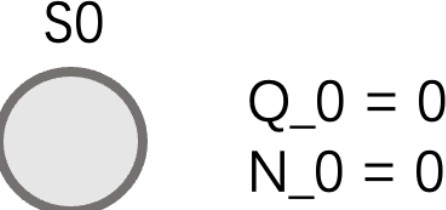

Figure I.1

**First iteration:** Node $S_0$ is the root and leaf node, and is not the terminating node, so it is extended. Assume that $S_0$ has two actions (the basic symbol [sin,x]) after it , which are transferred to $S_1$ and $S_2$ respectively.

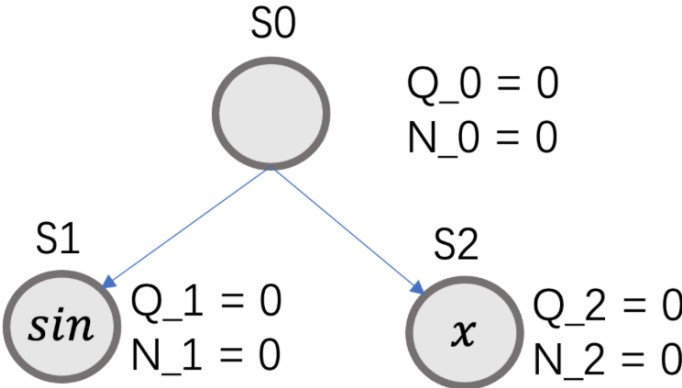

Figure I.2

You can then use the UCT formula to choose whether to extend $S_1$ or $S_2$. Here $N_1$ and $N_2$ are both 0, so the UCT value of both nodes is infinite, so any node can be selected, here $S_1$ is selected for extension and simulation (random selection of symbols). After simulation, it was found that the final reward value was 0.2, so it was updated retrospectively. $Q_1 = 0.2, N_1 = 1, Q_0 = 0.2, N_0 = 1$.

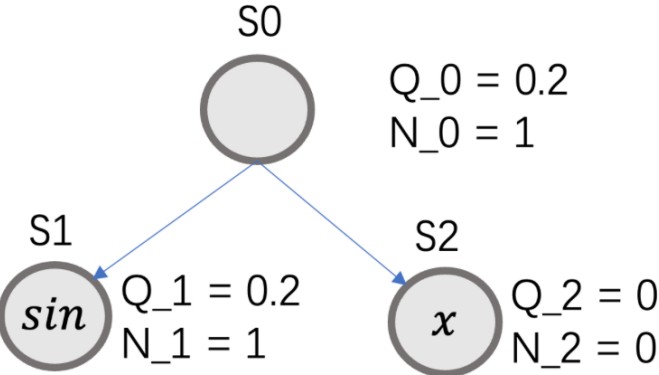

Figure I.3

**The second iteration:** Starting from $S_0$, calculate the UCT values of $S_1$ and $S_2$, and select the larger one for expansion. (assuming $S_1 > S_2$ after calculation)
Then according to the UCT value, $S_1$ is selected for expansion. After reaching $S_1$, it is found that it is a leaf node and has been explored, then enumerate all possible states of the current node (each action corresponds to a state), and add them to the tree.

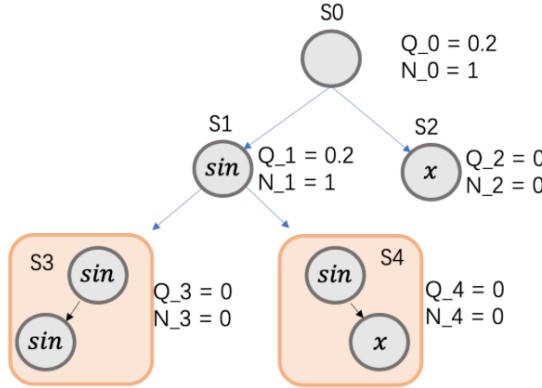

Figure I.4

Then we can select either $S_3$ or $S_4$ at random as before. Keep iterating. (In this example, S4 has successfully found the target expression)

