## A APPENDIX: PSEUDOCODE FOR THE ALPHASYMBOL

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

- Any unassigned operand is represented by , for instance, $^2$ corresponds to the square operator.

Table C.1: Symbol library and value range of the three data sets Nguyen, Korns, and Jin.

| Name | Expression | Dataset | Library |
|---|---|---|---|
| Nguyen-1 | $x_1^3 + x_1^2 + x_1$ | U$(-1, 1, 20)$ | |
| Nguyen-2 | $x_1^4 + x_1^3 + x_1^2 + x_1$ | U$(-1, 1, 20)$ | |
| Nguyen-3 | $x_1^5 + x_1^4 + x_1^3 + x_1^2 + x_1$ | U$(-1, 1, 20)$ | |
| Nguyen-4 | $x_1^6 + x_1^5 + x_1^4 + x_1^3 + x_1^2 + x_1$ | U$(-1, 1, 20)$ | |
| Nguyen-5 | $\sin(x_1^2)\cos(x) - 1$ | U$(-1, 1, 20)$ | |
| Nguyen-6 | $\sin(x_1) + \sin(x_1 + x_1^2)$ | U$(-1, 1, 20)$ | |
| Nguyen-7 | $\log(x_1 + 1) + \log(x_1^2 + 1)$ | U$(0, 2, 20)$ | |
| Nguyen-8 | $\sqrt{x}$ | U$(0, 4, 20)$ | |
| Nguyen-9 | $\sin(x) + \sin(x_2^2)$ | U$(0, 1, 20)$ | |
| Nguyen-10 | $2\sin(x)\cos(x_2)$ | U$(0, 1, 20)$ | |
| Nguyen-11 | $x_1^{x_2}$ | U$(0, 1, 20)$ | |
| Nguyen-12 | $x_1^4 - x_1^3 + \frac{1}{2}x_2^2 - x_2$ | U$(0, 1, 20)$ | |
| Nguyen-2$'$ | $4x_1^4 + 3x_1^3 + 2x_1^2 + x$ | U$(-1, 1, 20)$ | |
| Nguyen-5$'$ | $\sin(x_1^2)\cos(x) - 2$ | U$(-1, 1, 20)$ | |
| Nguyen-8$'$ | $\sqrt[3]{x}$ | U$(0, 4, 20)$ | |
| Nguyen-8$''$ | $\sqrt[3]{x_1^2}$ | U$(0, 4, 20)$ | |
| Nguyen-1$^c$ | $3.39x_1^3 + 2.12x_1^2 + 1.78x$ | U$(-1, 1, 20)$ | |
| Nguyen-5$^c$ | $\sin(x_1^2)\cos(x) - 0.75$ | U$(-1, 1, 20)$ | |
| Nguyen-7$^c$ | $\log(x + 1.4) + \log(x_1^2 + 1.3)$ | U$(0, 2, 20)$ | |
| Nguyen-8$^c$ | $\sqrt{1.23x}$ | U$(0, 4, 20)$ | |
| Nguyen-10$^c$ | $\sin(1.5x)\cos(0.5x_2)$ | U$(0, 1, 20)$ | |
| Korns-1 | $1.57 + 24.3 * x_1^4$ | U$(-1, 1, 20)$ | |
| Korns-2 | $0.23 + 14.2\frac{(x_4+x_1)}{(3x_2)}$ | U$(-1, 1, 20)$ | |
| Korns-3 | $4.9\frac{(x_2-x_1+\frac{x_1}{x_3})}{(3x_3))} - 5.41$ | U$(-1, 1, 20)$ | |
| Korns-4 | $0.13sin(x_1) - 2.3$ | U$(-1, 1, 20)$ | |
| Korns-5 | $3 + 2.13log(|x_5|)$ | U$(-1, 1, 20)$ | |
| Korns-6 | $1.3 + 0.13\sqrt{|x_1|}$ | U$(-1, 1, 20)$ | |
| Korns-7 | $2.1(1 - e^{-0.55x_1})$ | U$(-1, 1, 20)$ | |
| Korns-8 | $6.87 + 11\sqrt{|7.23x_1x_4x_5|}$ | U$(-1, 1, 20)$ | |
| Korns-9 | $12\sqrt{|4.2x_1x_2x_2|}$ | U$(-1, 1, 20)$ | |
| Korns-10 | $0.81 + 24.3\frac{2x_1+3x_2^2}{4x_3^3+5x_4^4}$ | U$(-1, 1, 20)$ | |
| Korns-11 | $6.87 + 11cos(7.23x_1^3)$ | U$(-1, 1, 20)$ | |
| Korns-12 | $2 - 2.1cos(9.8x_1^3)sin(1.3x_5)$ | U$(-1, 1, 20)$ | |
| Korns-13 | $32.0 - 3.0\frac{tan(x_1)}{tan(x_2)}\frac{tan(x_3)}{tan(x_4)}$ | U$(-1, 1, 20)$ | |
| Korns-14 | $22.0 - (4.2cos(x_1) - tan(x_2))\frac{tanh(x_3)}{sin(x_4)}$ | U$(-1, 1, 20)$ | |
| Korns-15 | $12.0 - \frac{6.0tan(x_1)}{e^{x_2}}(log(x_3) - tan(x_4))))$ | U$(-1, 1, 20)$ | |
| Jin-1 | $2.5x_1^4 - 1.3x_1^3 + 0.5x_2^2 - 1.7x_2$ | U$(-3, 3, 100)$ | |
| Jin-2 | $8.0x_1^2 + 8.0x_2^3 - 15.0$ | U$(-3, 3, 100)$ | |
| Jin-3 | $0.2x_1^3 + 0.5x_2^3 - 1.2x_2 - 0.5x_1$ | U$(-3, 3, 100)$ | |
| Jin-4 | $1.5\exp x + 5.0cos(x_2)$ | U$(-3, 3, 100)$ | |
| Jin-5 | $6.0sin(x_1)cos(x_2)$ | U$(-3, 3, 100)$ | |
| Jin-6 | $1.35x_1x_2 + 5.5sin((x_1 - 1.0)(x_2 - 1.0))$ | U$(-3, 3, 100)$ | |

Table C.2: Symbol library and value range of the three data sets neat, Keijzer and Livermore.

| Name | Expression | Dataset | Library |
|---|---|---|---|
| Neat-1 | $x_1^4 + x_1^3 + x_1^2 + x$ | $U(-1, 1, 20)$ | |
| Neat-2 | $x_1^5 + x_1^4 + x_1^3 + x_1^2 + x$ | $U(-1, 1, 20)$ | |
| Neat-3 | $\sin(x_1^2)\cos(x) - 1$ | $U(-1, 1, 20)$ | |
| Neat-4 | $\log(x + 1) + \log(x_1^2 + 1)$ | $U(0, 2, 20)$ | |
| Neat-5 | $2\sin(x)\cos(x_2)$ | $U(-1, 1, 100)$ | |
| Neat-6 | $\sum_{k=1}^{x} \frac{1}{k}$ | $E(1, 50, 50)$ | |
| Neat-7 | $2 - 2.1\cos(9.8x_1)\sin(1.3x_2)$ | $E(-50, 50, 10^5)$ | |
| Neat-8 | $\frac{e^{-(x_1)^2}}{1.2+(x_2-2.5)^2}$ | $U(0.3, 4, 100)$ | |
| Neat-9 | $\frac{1}{1+x_1^{-4}} + \frac{1}{1+x_2^{-4}}$ | $E(-5, 5, 21)$ | |
| Keijzer-1 | $0.3x_1 sin(2\pi x_1)$ | $U(-1, 1, 20)$ | |
| Keijzer-2 | $2.0x_1 sin(0.5\pi x_1)$ | $U(-1, 1, 20)$ | |
| Keijzer-3 | $0.92x_1 sin(2.41\pi x_1)$ | $U(-1, 1, 20)$ | |
| Keijzer-4 | $x_1^3 e^{-x_1} cos(x_1)sin(x_1)sin(x_1)^2 cos(x_1) - 1$ | $U(-1, 1, 20)$ | |
| Keijzer-5 | $3 + 2.13log(|x_5|)$ | $U(-1, 1, 20)$ | |
| Keijzer-6 | $\frac{x1(x1+1)}{2}$ | $U(-1, 1, 20)$ | |
| Keijzer-7 | $log(x_1)$ | $U(0, 1, 20)$ | |
| Keijzer-8 | $\sqrt{(x_1)}$ | $U(0, 1, 20)$ | |
| Keijzer-9 | $log(x_1 + \sqrt{x_1^2 + 1})$ | $U(-1, 1, 20)$ | |
| Keijzer-10 | $x_1^{x_2}$ | $U(-1, 1, 20)$ | |
| Keijzer-11 | $x_1 x_2 + sin((x_1 - 1)(x_2 - 1))$ | $U(-1, 1, 20)$ | |
| Keijzer-12 | $x_1^4 - x_1^3 + \frac{x_2^2}{2} - x_2$ | $U(-1, 1, 20)$ | |
| Keijzer-13 | $6sin(x_1)cos(x_2)$ | $U(-1, 1, 20)$ | |
| Keijzer-14 | $\frac{8}{2+x_1^2+x_2^2}$ | $U(-1, 1, 20)$ | |
| Keijzer-15 | $\frac{x_1^3}{5} + \frac{x_2^3}{2} - x_2 - x_1$ | $U(-1, 1, 20)$ | |
| Livermore-1 | $\frac{1}{3} + x_1 + sin(x_1^2))$ | $U(-3, 3, 100)$ | |
| Livermore-2 | $sin(x_1^2) * cos(x1) - 2$ | $U(-3, 3, 100)$ | |
| Livermore-3 | $sin(x_1^3) * cos(x_1^2)) - 1$ | $U(-3, 3, 100)$ | |
| Livermore-4 | $log(x_1 + 1) + log(x_1^2 + 1) + log(x_1)$ | $U(-3, 3, 100)$ | |
| Livermore-5 | $x_1^4 - x_1^3 + x_2^2 - x_2$ | $U(-3, 3, 100)$ | |
| Livermore-6 | $4x_1^4 + 3x_1^3 + 2x_1^2 + x_1$ | $U(-3, 3, 100)$ | |
| Livermore-7 | $\frac{(exp(x1) - exp(-x_1))}{2}$ | $U(-1, 1, 100)$ | |
| Livermore-8 | $\frac{(exp(x1) + exp(-x1))}{3}$ | $U(-3, 3, 100)$ | |
| Livermore-9 | $x_1^9 + x_1^8 + x_1^7 + x_1^6 + x_1^5 + x_1^4 + x_1^3 + x_1^2 + x_1$ | $U(-1, 1, 100)$ | |
| Livermore-10 | $6 * sin(x_1)cos(x_2)$ | $U(-3, 3, 100)$ | |
| Livermore-11 | $\frac{x_1^2 x_2^2}{(x_1+x_2)}$ | $U(-3, 3, 100)$ | |
| Livermore-12 | $\frac{x_1^5}{x_2^3}$ | $U(-3, 3, 100)$ | |
| Livermore-13 | $x_1^{\frac{1}{3}}$ | $U(-3, 3, 100)$ | |
| Livermore-14 | $x_1^3 + x_1^2 + x_1 + sin(x_1) + sin(x_2^2)$ | $U(-1, 1, 100)$ | |
| Livermore-15 | $x_1^{\frac{1}{5}}$ | $U(-3, 3, 100)$ | |
| Livermore-16 | $x_1^{\frac{2}{3}}$ | $U(-3, 3, 100)$ | |
| Livermore-17 | $4sin(x_1)cos(x_2)$ | $U(-3, 3, 100)$ | |
| Livermore-18 | $sin(x_1^2) * cos(x_1) - 5$ | $U(-3, 3, 100)$ | |
| Livermore-19 | $x_1^5 + x_1^4 + x_1^2 + x_1$ | $U(-3, 3, 100)$ | |
| Livermore-20 | $e^{(-x_1^2)}$ | $U(-3, 3, 100)$ | |
| Livermore-21 | $x_1^8 + x_1^7 + x_1^6 + x_1^5 + x_1^4 + x_1^3 + x_1^2 + x_1$ | $U(-1, 1, 20)$ | |
| Livermore-22 | $e^{(-0.5x_1^2)}$ | $U(-3, 3, 100)$ | |

Table C.3: Symbol library and value range of the three data sets Vladislavleva and others.

| Name | Expression | Dataset | Library |
|---|---|---|---|
| Vladislavleva-1 | $\frac{(e^{-(x1-1)^2})}{(1.2+(x2-2.5)^2))}$ | $U(-1,1,20)$ | |
| Vladislavleva-2 | $e^{-x_1}x_1^3 cos(x_1)sin(x_1)(cos(x_1)sin(x_1)^2-1)$ | $U(-1,1,20)$ | |
| Vladislavleva-3 | $e^{-x_1}x_1^3 cos(x_1)sin(x_1)(cos(x_1)sin(x_1)^2-1)(x_2-5)$ | $U(-1,1,20)$ | |
| Vladislavleva-4 | $\frac{10}{5+(x1-3)^2+(x_2-3)^2+(x_3-3)^2+(x_4-3)^2+(x_5-3)^2}$ | $U(0,2,20)$ | |
| Vladislavleva-5 | $30(x_1-1)\frac{x_3-1}{(x_1-10)}x_2^2$ | $U(-1,1,100)$ | |
| Vladislavleva-6 | $6sin(x_1)cos(x_2)$ | $E(1,50,50)$ | |
| Vladislavleva-7 | $2-2.1\cos(9.8x)\sin(1.3x_2)$ | $E(-50,50,10^5)$ | |
| Vladislavleva-8 | $\frac{e^{-(x-1)^2}}{1.2+(x_2-2.5)^2}$ | $U(0.3,4,100)$ | |
| Test-2 | $3.14*x1*x1$ | $U(-1,1,20)$ | |
| Const-Test-1 | $5*x1*x1$ | $U(-1,1,20)$ | |
| GrammarVAE-1 | $1./3+x1+sin(x_1^2))$ | $U(-1,1,20)$ | |
| Sine | $sin(x_1)+sin(x_1+x_1^2))$ | $U(-1,1,20)$ | |
| Nonic | $x_1^9+x_1^8+x_1^7+x_1^6+x_1^5+x_1^4+x_1^3+x_1^2+x_1$ | $U(-1,1,100)$ | |
| Pagie-1 | $\frac{1}{1+x_1^{-4}+\frac{1}{1+x2^{-4}}}$ | $E(1,50,50)$ | |
| Meier-3 | $\frac{x_1^2x_2^2}{(x_1+x_2)}$ | $E(-50,50,10^5)$ | |
| Meier-4 | $\frac{x_1^5}{x_2^3}$ | $U(0.3,4,100)$ | |