# OpenReview forum: "Discovering Mathematical Formulas from Data via LSTM-guided Monte Carlo Tree Search"
_ICLR.cc/2024/Conference — ICLR 2024 Conference Withdrawn Submission_

### Official Review · Reviewer_SBg1 · 2023-10-18

**Soundness:** 1 poor
**Presentation:** 1 poor
**Contribution:** 1 poor
**Rating:** 3
**Confidence:** 4

**Summary:**

This paper introduces a method to perform symbolic regression based on Monte-Carlo Tree Search (MCTS) with guidance from an LSTM.

**Strengths:**

I'm afraid I do not see much added value in this paper compared to existing work...

**Weaknesses:**

- Lack of novelty: the authors do not cite “Deep Generative Symbolic Regression with Monte-Carlo-Tree-Search” by Kamienny et al. The latter also performs MCTS for SR with guidance from a Transformer and should be cited — according to me, their method performs much better and is better validated empirically. There are also many other missing references in the SR literature.
- Experimental validation is poor. For example, the authors only report results when sampling only 20 points in the interval [-1,1], which is very small. They do not evaluate on the mainstream benchmark SRbench.
- Paper is particularly poorly written and presented, as detailed below.

**Questions:**

Important comments:
- I don’t understand what the \hat x_ij means in Eq 6; it does not seem to be defined anywhere. In symbolic regression, one typically predicts the labels \hat y from the inputs x_ij, but I don’t see how one can “predict the inputs”… This is very important as the authors consider this new loss function to be among their main contributions.

Comments on presentation:
- Many sentences are not capitalised
- Many sentences are cut with inappropriate punctuation (e.g. “which cleverly combines LSTM and MCTS. And outperforms several baselines” or “thereby avoiding situations where each symbol is predicted with a similar probability. Improved the search efficiency of the algorithm.”)
- References are not separated from the text with a space
- Fig 3 is poorly described: what is the red line in panel (a) ? What exactly is plotted in panel (c) (what is compressive strength) ?
- Lack of details in many parts:
    - “"No constrain" means no constraints are applied”, what are these constraints ?
    - Table 2 needs more details (“Yes/No”->”With entropy regularisation”/“Without”, “Time”->”Training time” etc)
Typos :
- “times it is child”
- “with the following expression:6”

Other things:
- “Anti-noise”->”demonising”
- “the algorithm’s reward function fluctuation is illustrated in the line graph (convergence proof)” : reward vs time is by no means a convergence proof…
- The computations after Eq. 7 do not make any sense : the partial derivatives are indicated as positive or negative without any justification on the range of the variables. Moreover, dy/dx>0 does not mean y is “proportional” to x.

---

> ### Author Response · Authors · 2023-11-12
> **Reply to the esteemed reviewer.**
>
> Dear reviewers, thank you very much for your review.
>
> **Weaknesses-1:**
>
> 1, There are essential differences between our algorithm and DGSR-MCTS in the way the policy network is trained. First, DGSR-MCTS trains the policy network inspired by the mutation in the GP algorithm, samples the expression as described in Figure 1 (by random mutation), and then predicts the following sequence with the previous sequence. A policy network is classically trained as a sequence-to-sequence encoder-decoder, using a cross-entropy loss. Instead, we train the policy network with the probability of $\pi$, normalized by the number of times MCTS visited each child node during the search (since MCTS tends to make nodes on the correct path more visited). We make the output of the policy network as close as possible to the learning outcome $\pi$ of MCTS; therefore, our training process is a numerical regression problem. And the updated policy network can give better guidance to MCTS. In our algorithm, the policy network and MCTS are coupled, helping each other and making progress together. Moreover, the training data of DGSR-MCTS is obtained using a similar random mutation, which has great randomness. However, our method is the real data obtained by MCTS through multiple simulations, which is more reliable and stable.
>
> 2,  Pretrained models are less general and are only good for variables and symbols that you have seen in the dataset. For example, if we only have [sin, cos] in our training set, it will not perform well when our symbol base becomes [sin, cos, exp]. When the variables are only $x_1$, and $x_2$, then it will be at a loss when encountering $x_3$, and $x_4$, and even when the sampling interval of variable X changes, it may affect the performance of the algorithm. However, our algorithm does not suffer from these problems, and our symbol library and the number of variables can be added or deleted at will. Moreover, X can be sampled in any interval.
> To sum up, we are very different from DGSR-MCTS, and our algorithm is more versatile and flexible than DGSR-MCTS. Moreover, our target data is more instructive in the training process of the policy network. In our algorithm, the policy network and MCTS are coupled together and make progress together in helping each other.
>
> We will cite DGSR-MCTS in the related work section of the paper, as well as add some SR articles. Thanks again for your review.
>
> **Weaknesses-2:**
>
> First of all, I am very sorry for the wrong expression of this sentence in our article. Thank you very much for your careful review and criticism. We will correct this sentence. We do not only sample 20 points on [-1,1]. Please refer to the appendix for the specific sampling interval and sampling number. Thanks again. As for the SRbench you said, first of all, SRbench mainly contains two parts of data, with concrete expressions and without concrete expressions. Since the evaluation criterion of our algorithm is the full recovery rate, it is not a simple comparison $R^2$. So we only selected the part of SRbench with concrete expressions (AIfeyman dataset). We have tested 222 mainstream formulas, which I think can reflect the performance of our algorithm comprehensively. Thank you very much again for your review.
>
> **Questions-1:**
>
> We propose a new loss function (Used to calculate rewards) called $S_{NRMSE}$. This innovation is rarely considered in previous papers. In the symbolic regression algorithm, when dealing with multivariate problems, if a variable is smaller than other variables, or the correlation between several variables is relatively high. At this time, the symbolic regression algorithm is prone to the problem of missing variables. Missing variables are not allowed because we want to fully recover the original expression. To solve this problem, we propose a new loss function $S_{NRMSE}$,
> $$
> \mathcal{S_{NRMSE}} = \frac{1}{\sigma_{y}}\sqrt{\frac{1}{\mathcal{N}}\sum_{i=1}^{\mathcal{N}} (y_{i} - \hat{y_i})^2 }  + \lambda \sum_{j = 1}^{m}\frac{1}{\sigma_{x_j}} \sqrt{\frac{1}{\mathcal{N}}\sum_{i=1}^{\mathcal{N}} (x_{ji}-\hat{x_{ji}})^2 }
> $$
> In $S_{NRMSE}$, instead of considering the difference between the true value y and the predicted value $y_{pred}$, we consider the difference between the true point and the predicted point in the geometric space. Suppose we have three variable $x_1, x_2, x_3 $, and forecasting formula is only $x_1, x_3 $so at this point, $\hat {x_1} = x_1, \hat {x_2} = 0, \hat{x_3} = x_3 $. In this case, the loss is large because of the lack of $x_2$. Where $x_{ji}$denotes the $i^{th}$ element of the $j^{th}$ variable.  We add the regulation coefficient $\lambda$ to the NRMSE between x and $x_{pred}$.  In the ablation experiments (Fig. 3(a)), we can see that $S_{NRMSE}$ can significantly improve the full recovery rate of the algorithm.
>
> **Questions-2:**
>
> Thank you very much for your careful reading and review. We will definitely improve the deficiencies mentioned in this part. Thanks.

---

> > ### Comment · Reviewer_SBg1 · 2023-11-21
> > **Response to rebuttal**
> >
> > Thank you for your response !
> > - W1: These are fair points, and I think it is important to add them to the paper. Thanks!
> > - W2: Indeed, this sentence is particularly misleading!
> > - Q1: Sorry but I still do not understand precisely what $\hat x$ means in this loss function. I strongly encourage the authors to formalize this better (the example with three variables is simply not clear enough -- what does "forecasting formula is $x1, x3$ mean? please use precise mathematical language), and add this to the manuscript.
> >
> > For now, I will increase my score up to 3. If the authors manage to explain better the loss function, I will consider bumping it up higher.

---

> > > ### Author Response · Authors · 2023-11-21
> > > **Answer the question of the esteemed reviewer on the loss function**
> > >
> > > Dear reviewers, thank you very much for your recognition of our work. We will write the above content in the paper according to your requirements. Thank you very much. Next, we will give you a detailed introduction of our proposed loss function $S_{NRMSE}$ by a concrete example.
> > >
> > > ​    Before I begin, let me answer one of your questions: "What $\hat{x}$ mean in this loss function?" In the same way that $\hat{y}$ is the predicted value of $y$, $\hat{x}$ is the predicted value of $x$. Specifically, $\hat{x}=x$ if the preorder traversal of the predicted expression contains the symbol $[x]$, and $\hat{x}=[0,0,...,0]$ if the preorder traversal of the predicted expression does not contain the symbol $[x]$. For example, suppose the preorder traversal of the target expression is:$[+, *,  x_1, x_2, x_3]$. However, the preorder traversal of the predicted expression is $[ *, x_2, x_3]$. Obviously, the predicted expression contains only the symbol $[x_2, x_3]$, not the symbol $[x_1]$. Thus, $\hat{x_2}=x_2$; $\hat{x_3}=x_3$;   $\hat {x_1} = [0, 0,..., 0] $. The reason for it equals $[0, 0, 0...0] $, in order to make it with $x_1 $have the bigger difference, thus have a great loss.
> > > ​     Next, we will expand the answer in detail with an example. First, the formula for $S_{NRMSE}$ is as follows. (We've changed the variable identity a bit for better understanding, specifically, '-pred' instead of '$\hat{\bullet}$' to represent the predicted value.) :
> > > $$
> > > \mathcal{S_{NRMSE}} = \frac{1}{\sigma_{y}}\sqrt{\frac{1}{\mathcal{N}}\sum_{i=1}^{\mathcal{N}} (y_{i} - y_{i-pred})^2 }  + \lambda \sum_{j = 1}^{m}\frac{1}{\sigma_{x_j}} \sqrt{\frac{1}{\mathcal{N}}\sum_{i=1}^{\mathcal{N}} (x_{ji}-x_{ji-pred})^2 }
> > > $$
> > > Suppose we have a data $[x_1, x_2, x_3, y] $, where $x_1 = [x_ {11}, x_ {12},..., x_{1n}]. x_2=[x_{21},x_{22},...,x_{2n}]; x_3=[x_{31},x_{32},...,x_{3n}]; $, $y = x_3 * x_2 + sin (x_1) $, the preorder traversal  is $[+, *,  x_3, x_2, sin, x_1] $, contains three variables in total. Suppose the expression predicted by our algorithm is $y_ {pred} = x_3 * x_2 $, the first sequence traversal is $[ *, x_2, x_3]$. At this point, the $y_ {pred} $ in only contains two of the three variables, $[x_3, x_2]$, lost the variable $[x_1]$. However, since the value of $sin(x_1)$ is much smaller than $x_3 * x_2$. So even though the predicted expression loses the variable $x_1$, the loss between $y$and $y_{pred}$ may be small if we calculate the loss using only the NRMSE loss function (first part of Eq. 1). This will give the algorithm the misleading impression that it has found the target expression when in fact it has not. So to solve the above problem, we propose the $S_{NRMSE}$ loss function. In $S_{NRMSE}$, we not only compute the loss between $y$ and $y_{pred}$(Eq. 1 part 1), **but also introduce the loss between $X$ and $X_{pred}$(Eq. 1 part 2).** I will focus on this part next. First of all, it is easy to understand that $ y$ is the actual value and $y_{pre}$ is the predicted value. Then similarly, $x_1$ is the true sampling value, and $x_{1-pred}$ is the predicted value; $x_2$ is the true sampled value, and $x_{2-pred}$ is the predicted value of $x_1$; $x_3$ is the true sampled value and $x_{3-pred}$ is the predicted value of $x_3$. Specifically, for the above example, the true expression is $y = x_3 * x_2 + sin(x_1)$; However, the predicted expression $y_{pred} = x_3 * x_2$. **Before we compute the loss, we already know what the prediction expression is and which symbols it contains. So we can know which variables are included in the predicted expression and which are missing.** **For example, in this case. We can know that the expression of forecast $y_ {pred} = x_3 * x_2 $ contains only two variables $[x_2, x_3] $, lost the variable $[x_1]$. So, we set $x_{2-pred}=x_2$; $x_{3-pred}=x_3$ ;  Due to the variable $x_1 $ lost, so we make $x_ {1 - pred} = [0, 0,..., 0] $, make it with a greater difference between the real $x_1 $.** In this case, for $x_2$, the prediction expression contains $x_2$, so its loss is as follows. $\frac{1}{\sigma_{x_2}}\sqrt{\frac{1}{\mathcal{N}}\sum_{i=1}^{\mathcal{N}}({x_{2i}-x_{2i-pred}})^2} =  \frac{1}{\sigma_{x_2}}\sqrt{\frac{1}{\mathcal{N}}\sum_{i=1}^{\mathcal{N}}({x_{2i}-x_{2i}})^2}=0$ ; For $x_3$, the prediction expression contains $x_3$, so the loss is:  $\frac{1}{\sigma_{x_3}}\sqrt{\frac{1}{\mathcal{N}}\sum_{i=1}^{\mathcal{N}}({x_{3i}-x_{3i-pred}})^2} =  \frac{1}{\sigma_{x_3}}\sqrt{\frac{1}{\mathcal{N}}\sum_{i=1}^{\mathcal{N}}({x_{3i}-x_{3i}})^2} = 0$ ; However, for the missing variable $x_1$, the loss is: $\frac{1}{\sigma_{x_1}}\sqrt{\frac{1}{\mathcal{N}}\sum_{i=1}^{\mathcal{N}}({x_{1i}-x_{1i-pred}})^2} =  \frac{1}{\sigma_{x_1}}\sqrt{\frac{1}{\mathcal{N}}\sum_{i=1}^{\mathcal{N}}({x_{1i}-0})^2} =  \frac{1}{\sigma_{x_1}}\sqrt{\frac{1}{\mathcal{N}}\sum_{i=1}^{\mathcal{N}}({x_{1i}})^2} \neq0$.

---

> > > > ### Author Response · Authors · 2023-11-21
> > > > **Supplement to the above answer**
> > > >
> > > > In this way, if the predicted expression has missing variables (certainly not fully recovering the target expression), then even though the loss between $y$ and $y_{pred}$ may be small, the missing variables produce a large loss.
> > > >
> > > > ​       **In short, if there are variables in the predicted expression, then we simply set the predicted value of these variables to their original value, and the loss is 0. If some variables are missing from the prediction expression, we set the predicted value of these variables to [0, 0,...,0] to generate a large loss. To achieve the purpose of preventing variable loss.**
> > > >
> > > > ​      Finally, it is my honor to answer your questions here. If you have any doubts, please do ask, we will be very happy to answer for you. Thank you very much again for your review.

---

> ### Comment · Reviewer_SBg1 · 2023-11-21
> **Answer**
>
> I'm sorry but this really does not make any sense to me.
> - Why would the penalty for missing out $x_i$ always be $\sum (x_i)^2$, independently of the particular way we missed out the variable $x_i$ ?! There are a vast number of ways one could "miss out" a variable, and the loss should be depend on this. For example, missing out a $sin(x_i)$ term is very different to missing out an $exp(-x_i)$ term;
> - Why would the penalty depend on the magnitude of $x_i$ ? Missing a term $sin(x_i)$ should give the same penalty, irrespectively of the scale of the $x_i$ sample during training;
> - What happens if on the contrary, one predicts a term $x_i$ which isn't in the ground truth ? This loss does not handle this case at all.
>
> In summary, this loss function is not principled at all in my opinion. I would like to hear the opinion of other reviewers or AC on this, but for now I will keep my score at 3.

---

> > ### Author Response · Authors · 2023-11-22
> > **Reply to the esteemed reviewer.**
> >
> > Dear reviewers, I am very sorry for not answering your questions. I will continue to answer for you in the following. Please give us another chance. Thank you.
> >
> > **Q1:**
> >
> > I think I need to explain to you the difference between missing variables and missing items. First, it should be stated that the missing variable $x_i$ is not the same as the missing term (for example, $sin(x_i)$, or $exp(x_i)$). The missing variable $x_i$ refers to whether the predicted expression contains the variable $x_i$(which can appear in any structure), and our method is designed for such cases. For example, for the formula $y = x_3 * x_2 + sin (x_1) $, the preorder traversal is $[+, *, x_3, x_2, sin, x_1] $, which contains three variables. If our predicted expression is $y_{pred}=x_1-x_2/x_3$,the preorder traversal is $[-,/,x_1,x_2,x_3]$, which has nothing to do with the true expression, but contains all three variables without missing any variables, So its loss on $X$ (the second term in Eq. 1) is 0, but its loss on $Y$(the first term in Eq. 1) will be large.
> >
> > If we lose the variable $x_i$, then we have a term $\sum(x_i)^2$ in our loss, which is the extra loss after we lose the variable. As for the "missing out a $sin(x_i)$term is very different to missing out an $exp(x_i)$term" you mentioned, in this case, due to the different range of the two, the loss between $y$and $y_{pred}$ will also be very different. The magnitude of this loss is reflected by $\frac{1}{\sigma_{y}}\sqrt{\frac{1}{\mathcal{N}}\sum_{i=1}^{\mathcal{N}}(y_i-y_{perd})^2}$(the first term of Eq. 1). More precisely, since the range of $sin(x_i)$ is relatively small, the loss of $y$and $y_{pred}$(the first term in Eq. 1) generated by losing $sin(x_i)$ will be relatively small. However, for the loss of $exp(x_i)$, the loss of $y$and $y_{pred}$(the first term of Eq. 1) will be larger.
> >
> > Thus, losing $sin(x_i)$ and $exp(x_i)$will yield completely different loss values for our loss function. But the difference is mainly reflected in the loss of $ y$ and $y_{pred}$(the first term of Eq. 1).
> >
> > **Q2:**
> >
> > Dear reviewers. The two parts of the loss function have their own tasks, the second part is only responsible for checking whether the variable is missing, if the variable $x_i$ is missing, then there is an additional penalty $\sum(x_i)^2$. As for whether you said to lose $sin(x_1)$or $exp(x_1)$, these will make the difference between $y$ and $y_{pred}$, which will be reflected in the first part of the loss. Thank you for your review.
> >
> > **Q3**
> >
> > Dear reviewer, in the problem of symbolic regression, we will know that there are a total of several variables in the data before the prediction begins. Generally, this situation will not occur.  And even if what you said happened, since we are imposing a soft penalty on X, we can control the weight of the second part of the loss by tuning the hyperparameter $\lambda$ in Eq. 1. If $\lambda=0$ then it degenerates to NRMSE loss. In this way, even if some unnecessary variables are lost, the impact on the overall loss is small due to the small proportion of the second part loss. At this time, if the first part loss is very low, we will judge that the target expression has been found. Impose a soft penalty on $X$. This is a flexible place for $S_{NRMSE}$.

---

### Official Review · Reviewer_YgjN · 2023-10-28

**Soundness:** 2 fair
**Presentation:** 1 poor
**Contribution:** 2 fair
**Rating:** 3
**Confidence:** 4

**Summary:**

The submission examines the performance of an AlphaZero-like approach, which they call AlphaSymbol, to the symbolic regression problem.

**Strengths:**

I'm not very familiar with the symbolic regression, but I'm not aware of AlphaZero having been applied to this problem setting.

**Weaknesses:**

The formatting of the submission makes it hard to read. There is not sufficient space between the paragraphs. There is clearly content that can be cut from the submission to make it easier to read. For example, the four phases of MCTS do not need to be enumerated in the introduction.

The structure and contextualization of the submission is poor. The submission is essentially applying AlphaZero to a new setting with problem-specific tweaks. However, the submission is written as if the AlphaZero methodology is largely original to the submission: AlphaGo Zero is cited one time for the definition of a running action value and AlphaZero is not cited at all. This lack of proper attribution is alone enough to disqualify the submission from acceptance. The appropriate way to structure the submission would be to include AlphaZero in a background section and describe problem specific tweaks in a methodology section.

There are also some strange deviations from AlphaZero that make me skeptical of whether the results should be taken seriously. For example, in equation (4), the submission seems to suggest that it uses the normalized logarithm of the visit counts as the policy (though it gives contradictory information elsewhere in the submission). If it is true that the submission is using the logarithm of the visit counts, it ought to better justify this modification (though I am skeptical that a justification exists). Also, it adds an entropy penalty to the loss function that is not typically present. The submission does ablations which seem to suggest that this entropy penalty is helpful. But these lead me to wonder whether this penalty is only necessary because of other unusual choices made by the submission. Overall, it is certainly possible that the submission's deviations from AlphaZero are necessary to achieve good performance, but the submission's poor presentation leaves the reader with the feeling that these deviations are haphazard rather than the product of careful study.

**Questions:**

> Think of the things where a response from the author can change your opinion

I think the submission requires significant revisions to improve readability, appropriately separate background from contribution, and discuss and investigate the reasoning behind deviations from AlphaZero.

---

> ### Author Response · Authors · 2023-11-12
> **Reply to the esteemed reviewer**
>
> Dear reviewers, thank you very much for taking time out of your busy schedule to review my article.
> **Weaknesses-1:**
>
> Dear reviewers, we will carefully cut and improve our article according to your requirements. The four stages of MCTS are included in the introduction so that readers who are not familiar with the MCTS process can better understand the article.
>
> **Weaknesses-2:**
>
> First of all, I apologize to the authors of AlphaZero[1], and to you, for forgetting to cite them in our article. We do not mean not to quote, please trust us. It would be extremely unwise for us to hide the fact that we are borrowing from AlphaZero by not citing its well-known work. Furthermore, our algorithm is called Alphasymbol to remind you of our connection to the Alpha family of algorithms. So trust us, we're not intentionally not citing AlphaZero articles. We will definitely clarify the relationship between AlphaZero and our algorithm in the final submission as per your request. Thanks again for your review and suggestions.
>
> [1] : Silver D, Hubert T, Schrittwieser J, et al. Mastering chess and shogi by self-play with a general reinforcement learning algorithm[J]. arXiv preprint arXiv:1712.01815, 2017.
>
> **Weaknesses-3:**
>
> Dear reviewers. We just use the log normalization of the access count in the self-search phase as the basis for selecting the next symbol. However, in the MCTS search process, we apply the UCT in Equation 1 to select the next node symbol.
>
> In AlphaGo zero the article [1], $\ pi_ {a_i} = N (s, a_i) ^ {1/ \tau} / (\sum_0 ^ N (N (s, a_j) ^ {1/ \tau})) $. In Eq. 4, we use a small trick by taking log first and then normalizing it. The advantage is that the data can be transformed into a logarithmic scale to prevent the N value from fluctuating too much and affecting the training stability of the LSTM.
>
> For the problem of entropy loss, we want LSTM to provide effective guidance for MCTS. Then we want the LSTM to be as "confident" as possible. For example, for the following two probability distributions, [0.25, 0.25, 0.3, 0.2] and [0.1, 0.1,0.7, 0.1], although both of these probability distributions ultimately choose the third symbol, we would prefer that the LSTM output the second probability distribution. Because it's more instructive. This operation can be proved to be effective by ablation experiments and Table 2. Thank you very much for your review.
>
> [1] : Silver, D., Schrittwieser, J., Simonyan, K. *et al.* Mastering the game of Go without human knowledge. *Nature* **550**, 354–359 (2017). https://doi.org/10.1038/nature24270
>
> **Questions**
>
> Dear reviewers, in the final version, we will carefully revise the article according to your and other reviewers' opinions, so as to improve the readability of the article. Once again, thank you for reviewing our article and wish you a happy life.

---

> > ### Author Response · Authors · 2023-11-21
> > **Ask a respected reviewer for advice.**
> >
> > Dear reviewers, I do not know if you have any questions about our paper. If so, I sincerely hope you can raise them here, and we will be happy to answer your questions. Thank you again for taking time out of your busy schedule to review our manuscript. Wish you a happy life.

---

> > > ### Comment · Reviewer_YgjN · 2023-11-21
> > > **Response**
> > >
> > > Thanks for the reply. I think I would've needed to see a revised version of the submission to have considered the possibility of changing my score.

---

> > > > ### Author Response · Authors · 2023-11-21
> > > > **Reply to the esteemed reviewer**
> > > >
> > > > Dear reviewers, thank you very much for your review of our paper. In the revised version of the paper, we have made the following modifications according to your requirements.
> > > > 1. Deleted some redundant parts in the article, such as the four steps of MCTS.
> > > > 2, show the relationship between our work and AlphaZero in the abstract and introduction, and cite AlphaZero articles. Once again, apologies to you and the AlphaZero authors
> > > > 3. Redraw the pictures in the paper to make them contain more information and more accurate representation.
> > > > 4. some statements in the article were polished.
> > > > Finally, if you think our paper needs to be modified, please continue to give us suggestions. We will be more than happy to improve our paper according to your suggestions. Once again, thank you for your review, and wish you a happy life.

---

### Official Review · Reviewer_Fmza · 2023-10-31

**Soundness:** 3 good
**Presentation:** 3 good
**Contribution:** 2 fair
**Rating:** 6
**Confidence:** 5

**Summary:**

The paper considers using a Monte Carlo Tree Search variant for discovering mathematical formulas. The MCTS variant uses PUCT for selection with an LSTM network providing the prior.
The empirical evaluation show that the algorithm is competitive with the state-of-the-art on several benchmarks.

**Strengths:**

The empirical results do show that the proposed algorithm can be a powerful tool for discovering mathematical formulas.

**Weaknesses:**

The proposed algorithm is a fairly standard MCTS, LSTM being the only slight deviation from a standard architecture used in games.

**Questions:**

Since the main deviation from the standard MCTS implementation (that uses deep neural networks as priors) is the use of LSTM, it would have been useful to explore the possible alternative architectures. LSTM seems a reasonable choice given previous suitability to formula discovery, but have you tested other architectures as well?

---

> ### Author Response · Authors · 2023-11-12
> **Reply to the esteemed reviewers**
>
> Dear reviewers, thank you very much for your careful and conscientious review.
>
> In theory, other neural networks that process sequences can be used as well, but the big advantage of LSTM is that it has a "forget gate", which automatically selects the information to memorize during the learning process. In this way, when we only input the parents and siblings of the node to be predicted, the LSTM can remember the important information and discard the unimportant information from time to time. However, like Transformers, we might have to input all the nodes before the node we want to predict each time.  such as for sequence [1,2,3,4,5,6], for LSTM, we can enter [1,2] to predict [3], then [2,3] to predict [4]...  Although each input is a part of the sequence, the LSTM automatically remembers important information from each previous input sequence. With the transformer, we need [1,2] to predict [3], and then [1,2,3] to predict [4]...  Each time, you enter all the previous nodes.
>
> In summary, other sequence prediction algorithms can also be theoretically used for our algorithm, but we think LSTM may be more suitable for some.
>
> Once again, thank you very much for your careful review. Thank you very much. I wish you every success in your work.

---

> > ### Comment · Reviewer_Fmza · 2023-11-22
> >
> > Indeed LSTM seems a reasonable choice, but my question was about testing. It seems not, but it was not a major issue.

---

> > > ### Author Response · Authors · 2023-11-22
> > > **Thanks to the esteemed reviewer**
> > >
> > > Dear reviewer, thank you very much for your careful review of our work. We are honored to have our work reviewed by you. Thank you once again and wish you a happy life.

---

### Official Review · Reviewer_27mA · 2023-11-02

**Soundness:** 3 good
**Presentation:** 2 fair
**Contribution:** 3 good
**Rating:** 5
**Confidence:** 3

**Summary:**

The paper presents AlphaSymbol, a new approach for symbolic regression for the discovery of mathematical formulae. The proposed approach augments a monte-carlo tree search with an LSTM to guide the search, a new reward function that addresses the problem of variable omission, and a new loss function for training the LSTM such that it produces distributions with lower information entropy. The experiments show that the proposed approach has a significantly higher recovery rate compared to the baselines.

**Strengths:**

Strengths:
- Important and well-motivated problem (symbolic regression for discovering mathematical formulae)
- New approach for the problem that consists of using LSTM to guide the monte-carlo tree search, as well as using a specialized reward function and a specialized loss function for training the LSTM
- Experiments show significantly higher recovery rate compared to the baselines

**Weaknesses:**

Weaknesses:
- Evaluation of experiments is not entirely clear: When is the search stopped and counted as "not recovered"?
- No comparison of running times between the proposed approach and the baselines. Or alternatively, comparison of rewards over time vs. the baselines.
- Some details about the technical approach is not entirely clear:
	* It is not clear how is the self-search phase and the use of LSTM are coordinated. For example, is the self-search used for several epochs while LSTM is being trained and then the algorithm changes to using the trained LSTM (if so, when is the change done)?
	* There are two loss functions. Is the second one (S_{NRMSE}) only used for the reward computation (while the first one is used for the LSTM training)?
- Writing can be improved as some details are missing (examples above), format is quite dense with some subtitles appear inside a paragraph (e.g., "Ablation experiment for information entropy."), and several typos and inconsistencies (examples listed under "Minor issues" below). The appendix is used as part of the paper, simply transferring some figures there and referencing to them as if they are part of the main paper, which hinders the ability of the paper itself to be self-contained without the appendix and hurts the readability of the paper. Section 5 is entitled "Discussion" but reads much more like a "Conclusion".


Minor issues:
- in abstract: "MCTS and LSTM hand in hand advance together, win-win cooperation until the target expression is successfully determined" - this is a bit too informal and can be rephrased to be a bit more precise/clear.
- "which is not interpretable and analyzable": there are many post-hoc interpretation techniques that can be applied
- " visit count N increase": what is N?
- "regression. however" -> "regression. However"
- Section 4: the description of algorithms as "excellent", "superior" is not clear (is excellent better than superior?). It is also important to highlight the current state-of-the-art on this task.
- "method. the expression 5 shows" -> "method. Expression 5 shows"
- "matrixE.1,"

**Questions:**

Please see "weaknesses" above.

---

> ### Author Response · Authors · 2023-11-12
> **Reply to the esteemed reviewers**
>
> Dear reviewers, thank you very much for taking time out of your busy schedule to review our manuscript.  I wish you every success in your work.
>
> **Weaknesses-1:**
>
> Dear reviewers, we use "full recovery rate" in our paper. For example, for the expression $sin(x) + x$, the formula binary tree is expanded to [+, sin, x, x] according to the preorder traversal, that is to say, the algorithm must obtain an expression that is the same as $sin(x) + x$to be restored. That is, the sequence [+,sin,x,x] is generated one by one. Secondly, in the algorithm, we will specify a threshold of the maximum number of expressions searched in the self-search phase, for example, 1000. If the target expression is not found after finding 1000 expressions in the self-search phase, the search will stop and it will be regarded as failing to recover the expression.
>
> **Weaknesses-2:**
>
> Dear reviewers, due to time reasons, we only test the running time (in seconds) of each algorithm on the Nguyen dataset, run each expression 3 times, and then take the average, and the results are as follows.
>
> | Dataset   | AlphaSymbol | DSO     | SPL    | GP      |
> | --------- | ----------- | ------- | ------ | ------- |
> | Nguyen-1  | 14.22       | 7.05    | 18.29  | 87.55   |
> | Nguyen-2  | 115.34      | 96.59   | 188.23 | 236.27  |
> | Nguyen-3  | 132.64      | 108.88  | 316.35 | 443.20  |
> | Nguyen-4  | 268.42      | 222.41  | 589.54 | 857.35  |
> | Nguyen-5  | 624.45      | 1647.05 | 831.28 | 1087.55 |
> | Nguyen-6  | 136.24      | 99.79   | 174.61 | 236.27  |
> | Nguyen-7  | 36.24       | 808.38  | 84.99  | 1443.20 |
> | Nguyen-8  | 1.22        | 3.41    | 1.23   | 3.35    |
> | Nguyen-9  | 17.18       | 13.05   | 24.14  | 87.55   |
> | Nguyen-10 | 38.28       | 1299.79 | 53.24  | 1236.27 |
> | Nguyen-11 | 62.44       | 108.88  | 80.73  | 443.20  |
> | Nguyen-12 | 323.42      | 2857.35 | 487.90 | 2444.63 |
> | Average   | 147.5       | 606.05  | 237.54 | 717.23  |
>
> **Weaknesses-3-1:**
>
> First of all, we will perform multiple MCTS simulations in the self-search phase. For example, we have a total of three symbols [sin,cos,x], the currently selected symbol is [sin], and we want to predict the next symbol. After multiple simulations with MCTS (with LSTM guidance), of the three children of [sin], x has been visited 8 times, sin has been visited once, and cos has been visited once. At this point, we know that sin followed by x is probably the right choice, so we need to teach the LSTM to remember this lesson. So let's say $\pi = softmax([1,1,8])$. The LSTM is trained with [sin] as input so that its output is closer to $\pi$. The trained LSTM is then used to better guide MCTS to produce better $\pi$ to train a more powerful LSTM network. This process continues until the end. During the self-search process, we will put {[sin], $\pi$} training data like this in a library, the capacity of the library is [1000], if more than 1000, new in, old out. We take a batch of data from the library at a time to train the LSTM. As for when to replace the LSTM, after training the LSTM for a certain number of times, we will test the LSTM, specifically, let the LSTM with the new parameters guide MCTS to simulate multiple expressions, and if the maximum reward of the obtained expression is higher than that of the previous simulation, we will replace the old LSTM parameters.
>
> **Weaknesses-3-2:**
>
> Dear reviewers, you are right, the first loss function is used to train LSTM; The second loss function, $S_{NRMSE}$, is used to compute the reward after the search yields an expression.  We will improve this in this article to make it more clear. Thank you again
>
> **Weaknesses-4:**
>
> Dear reviewers, thank you very much for your careful review. In the final submission version, we will carefully improve the article one by one according to your requirements. Thank you very much for your valuable advice.

---

> > ### Comment · Reviewer_27mA · 2023-11-23
> >
> > Thank you for your comment. The results on runtime are interesting but should be provided for the other datasets. I think the processes described under Weaknesses-1 and Weaknesses-3-1 need to be explained more clearly in the paper and the formatting of the paper should be improved as mentioned before and by the other reviewers as well.

---

> ### Author Response · Authors · 2023-11-21
> **Ask a respected reviewer for advice.**
>
> Dear reviewers, I do not know if you have any questions about our paper. If so, I sincerely hope you can raise them here, and we will be happy to answer your questions. Thank you again for taking time out of your busy schedule to review our manuscript. Wish you a happy life.

---

### Author Response · Authors · 2023-11-21
**Report current improvements to the paper to the respected reviewers**

Dear reviewers, thank you very much for taking time out of your busy schedule to review my paper. I have revised the paper according to the requirements of teachers, and I have uploaded the revised manuscript again and marked the revised part in red. The specific changes are as follows:
1. Newly wrote the abstract and introduction of the paper, specifically, added some new references, and made a detailed description of the advantages of our work over the existing algorithms in the paper.
2. Corrected some improper words or expressions in the paper, and corrected some mistakes.
3. Redraw the first three figures of the paper to make them more understandable, and modify some errors.
4. The logic of some paragraphs in the paper was properly modified to make it easier to read.

Finally, if you have any questions about our work, please feel free to ask them here. We will be happy to answer your questions. Once again, I would like to thank all the reviewers for reviewing our manuscript and wish you a happy life.

---

### Public Comment · ~Chandan_K._Reddy1 · 2023-11-21
**Suggestions on improvement**

This paper employs Monte Carlo tree search with LSTM-based priors for symbolic regression optimization. The authors should look into the following suggestions on some important comparisons:

1. The paper should consider reporting the results on the SRBench, a well-accepted benchmark in the symbolic regression community that features complex, higher-dimensional problems. This benchmark's inclusion would enhance the evaluation, especially given its comparison with 14 superior symbolic regression baselines not considered in this study.
2. A recent NeurIPS 2023 paper "Transformer-based Planning for Symbolic Regression" (https://arxiv.org/abs/2303.06833) introduced a highly similar approach using MCTS lookahead planning on top of pre-trained Transformers symbolic regression models (instead of LSTM policy suggested here). I believe a comparison to this baseline will strengthen the work, especially given the similarities of the underlying key ideas. More intuition on why an LSTM-based approach is better than a transformer-based model can be helpful.

---

> ### Author Response · Authors · 2023-11-22
> **Response to the reviewer**
>
> Dear teacher, thank you very much for your concern and support for our work. I will respond to your questions.
>
> **Q1**
>
> Dear teacher, I am glad to answer your question. As for the SRbench problem you mentioned, first of all, SRbench mainly contains two parts of data, with concrete expressions and without concrete expressions. Since our algorithm is judged by the full recovery rate ( the goal expression is fully recovered，$R^2=1$), it is not a simple comparison of $R^2$. While $R^2 > 0.99$ can be considered to recover the formula, we believe that the full recovery rate is a more challenging task and a better indicator of the performance of the algorithm. In view of this, we chose the part of SRbench with concrete expressions (AIfeyman dataset), so we are not completely untested on the SRbatch dataset. And we have tested on 222 mainstream formulas, which I think can reflect the performance of our algorithm comprehensively. Thank you very much again for your review.
>
> **Q2**
>
> Dear teacher, we have introduced the advantages and disadvantages of TPSR compared with our algorithm in the introduction part of the paper in detail. I'll answer your questions next. In our algorithm, the policy network and MCTS are coupled, helping each other and making progress together.  Our policy network is trained and improved in real time using real data obtained from multiple simulations of MCTS. However, TPSR directly uses an end-to-end large-scale pre-trained model as its policy network, and its parameters are not updated during the whole process.
> Furthermore, as a pre-trained model, TPSR is less general and is only good for variables and symbols that you have seen in the dataset. For example, if we only have $[sin.cos]$ in our training set, it will not perform well when our symbol base becomes $[sin, cos, exp]$. When the variables are only $x_1$,$x_2$, then it will be at a loss when encountering $x_3$,$x_4$, and even when the sampling interval of variable x changes, it may affect the performance of the algorithm. However, our algorithm does not suffer from these problems, and our symbol library and the number of variables can be added or deleted at will. Moreover, $X$ can be sampled in any interval.
>
>  To sum up, we are very different from TPSR, and compared with TPSR, our algorithm has more advantages in generality and flexibility.